# Genome-wide quantification of contributions to sexual fitness identifies genes required for spore viability and health in fission yeast

R. Blake Billmyre[1][☯], Michael T. Eickbush[1][☯], Caroline J. Craig[1], Jeffrey J. Lange[1], Christopher Wood[1], Rachel M. Helston[1], Sarah E. Zanders[1,2]*

**1** Stowers Institute for Medical Research, Kansas City, Missouri, United States of America, **2** Department of Molecular and Integrative Physiology, University of Kansas Medical Center, Kansas City, Kansas, United States of America

☯ These authors contributed equally to this work.
* sez@stowers.org

**Data Availability Statement:** Next-generation sequence data is available at the NCBI SRA under accession number PRJNA758956. All other

## Abstract

Numerous genes required for sexual reproduction remain to be identified even in simple model species like *Schizosaccharomyces pombe*. To address this, we developed an assay in *S. pombe* that couples transposon mutagenesis with high-throughput sequencing (TN-seq) to quantitatively measure the fitness contribution of nonessential genes across the genome to sexual reproduction. This approach identified 532 genes that contribute to sex, including more than 200 that were not previously annotated to be involved in the process, of which more than 150 have orthologs in vertebrates. Among our verified hits was an uncharacterized gene, *ifs1* (**i**mportant **f**or **s**ex), that is required for spore viability. In two other hits, *plb1* and *alg9*, we observed a novel mutant phenotype of poor spore health wherein viable spores are produced, but the spores exhibit low fitness and are rapidly outcompeted by wild type. Finally, we fortuitously discovered that a gene previously thought to be essential, *sdg1* (**s**ocial **d**istancing **g**ene), is instead required for growth at low cell densities and can be rescued by conditioned medium. Our assay will be valuable in further studies of sexual reproduction in *S. pombe* and identifies multiple candidate genes that could contribute to sexual reproduction in other eukaryotes, including humans.

## Author summary

Sex is absolutely required for many organisms, including humans, to reproduce. However, we still lack a complete understanding of the genetic contributions to sexual reproduction, even in many model organisms. Here we use a high-throughput insertional mutagenesis approach to measure the consequences during sexual reproduction of gene disruption across the genome in fission yeast. As a result, we identified hundreds of novel genes as contributing to sexual reproduction. While we identified a mutant with the expected loss of spore viability phenotype, we also demonstrated that disruption of some of these genes resulted in viable but low-quality spores (analogous to gametes like sperm in humans). In

relevant data are within the manuscript and its Supporting Information files or at the Stowers original data repository (https://www.stowers.org/research/publications/libpb-1654). Instructions for accessing data from the Stowers original data repository via FTP can be found at https://www.stowers.org/research/publications/odr#ftp.

**Funding:** This work was funded by K99/R00 funding to SEZ (GM114436) and DP2 funding to SEZ (GM132936) from the NIGMS, by a Searle Scholars Award to SEZ, and by the Stowers Institute for Medical Research to SEZ. The funders has no role in study design, data collection and analysis, decision to publish, or preparation of the manuscript. All authors received salary support from the Stowers Institute for Medical Research.

**Competing interests:** I have read the journal's policy and the authors of this manuscript have the following competing interests: SEZ: Inventor on patent application 834 serial 62/491,107 based on wtf killers.

addition, we also identified a gene that was not required for sex itself but was instead required for cells to grow at low density. We propose that this mutant is a representative of a new class of genes that we refer to as "social distancing genes" because they are unable to grow without the presence of neighbors. In sum, this work presents genome-wide measurement of the genetic contributions to sex in fission yeast.

## Introduction

Sexual reproduction requires diploid cells to generate haploid gametes. Compatible gametes can then fuse to regenerate the diploid state. As a result, sexual reproduction produces recombinant offspring with the same number of chromosomes as the parents. Sexual reproduction is conserved broadly across the eukaryotic kingdom, with few exceptions [1], and thus is inferred to have been present in the common ancestor of all eukaryotes [2,3]. Long-lived asexual eukaryotic lineages are rare, suggesting that loss of sex may produce lineages that can be capable of short-term success but are unable to adapt and persist over long evolutionary time scales, making asexual eukaryotic lineages evolutionary dead-ends [3]. Despite the ubiquity of sexual reproduction, the machinery required for meiosis and sexual reproduction is not fully understood even in well-studied model organisms.

Much of our understanding of the mechanisms underlying sexual reproduction was derived from classic forward genetic screens (including but not limited to [4–13]). These screens generate enormous progress in biology but are often technically challenging because of the effort required to isolate, screen, and map the mutants identified. Forward genetic screens also run the risk of incomplete saturation and thus false negatives on a whole-genome search for genes involved in a process. Subsequently, reverse genetic screens, often of deletion collections in well-studied model organisms, have provided substantial additional insight into the meiotic toolkit of individual species (including but not limited to [14–21]). Deletion collection screens are powerful, but their use is typically limited to organisms with large research communities capable of collaboratively constructing these deletion collections. In contrast, non-model organisms rarely have these resources available, and construction of these deletion collections is cost and time prohibitive. In addition, many existing deletion collections are found in organisms that are relatively distantly related, often on the order of hundreds of millions of years diverged [22,23]. As a result, studying the evolution of rapidly evolving processes like sexual reproduction is difficult using these existing sets of tools and approaches.

An alternative high-throughput and whole genome approach is Transposon insertion sequencing (TN-seq) [24–27]. TN-seq works by generating massive, pooled libraries of haploid cells wherein each cell contains a single transposon insertion at a random site in the genome. These insertion sites are then specifically mapped by sequencing, often including PCR amplification of the insertion site. With sufficiently dense transposon insertions across the genome, essential genes or regions can be identified after sequence analysis as regions with significantly diminished transposon insert density relative to the rest of the genome (Fig 1A and 1B). Subsequent experiments can apply a selective pressure to this mutant library and then resequence to identify changes in insert frequencies from the first growth condition to the second (Fig 1C). This approach allows quantitative and comparative pooled analysis of the equivalent of a signature tagged whole genome deletion collection in a single experiment without the need to construct a deletion collection. In addition, the vast increase in mutant density per gene in a TN-seq library enables more accurate estimates of the contribution of individual genes to a phenotype of interest. TN-seq reduces the scale of whole genome reverse genetic screens to one that

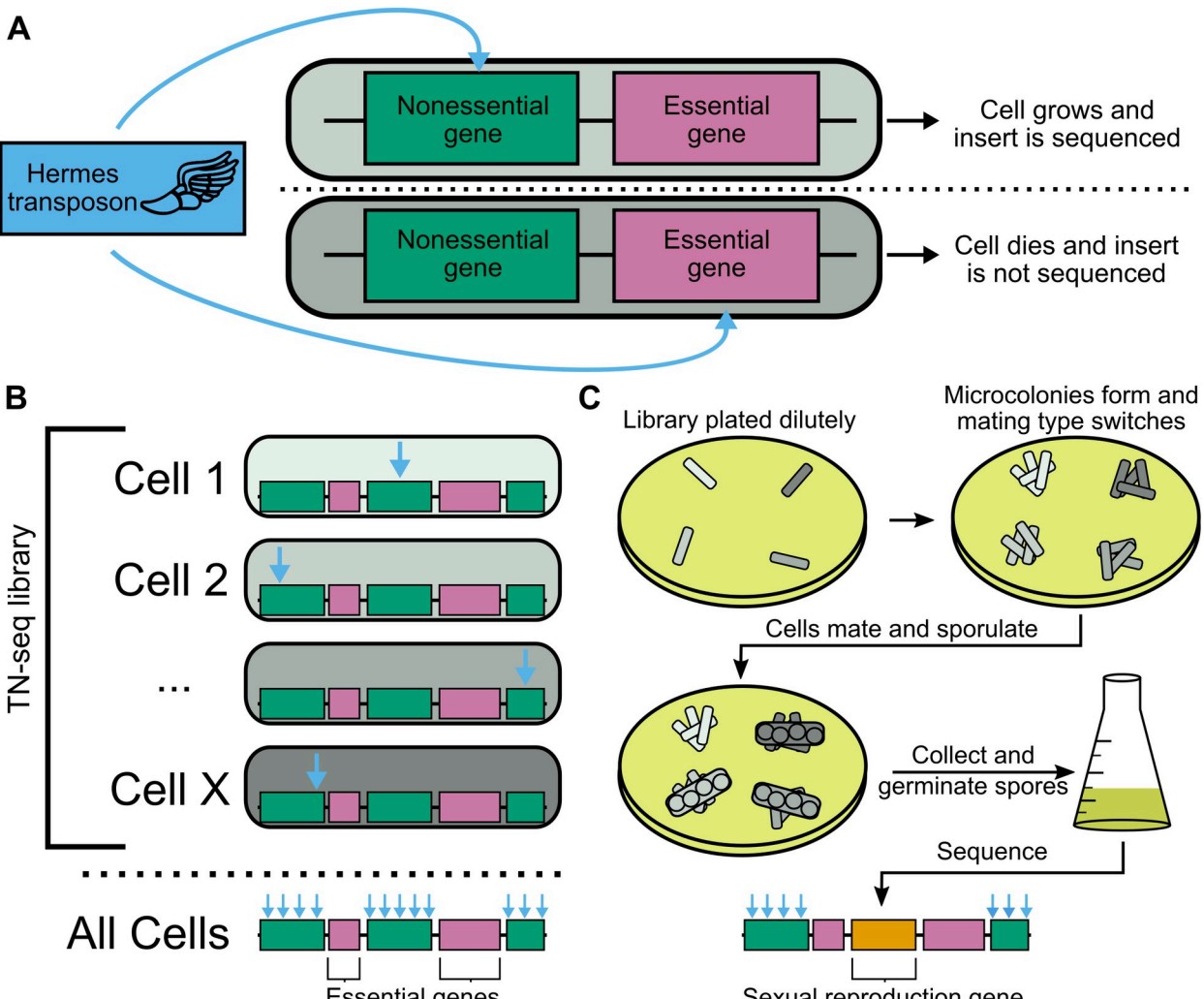

**Fig 1. Transposon insertion sequencing can identify sexual reproduction genes.** A) Transposon insertions will affect cells differently depending on where they occur in the genome. Inserts into nonessential genes (top) are unlikely to be strongly deleterious and thus will be recovered and sequenced. In contrast, insertions into essential genes (bottom) are likely to kill the cell and not be available to sequence. B) Building a library of cells with independent transposon insertions allows mapping of essential (pink) and non-essential (green) regions of the genome by measuring insertion density via sequencing. C) A transposon insertion library can be subsequently selected for the ability to undergo sexual reproduction to produce viable spores. The library is plated at low density on MEA plates so that clonal microcolonies form. The cells within the microcolony can undergo mating type switching and mate with siblings to form diploids homozygous for a transposon insertion. The diploids then undergo meiosis and sporulation. We then collect spores, germinate them in liquid medium, and assay insert frequency and location via sequencing. Genes with a role in sexual reproduction that previously tolerated insertions will no longer have inserts (orange).

can be accomplished within a single lab, rather than an entire community working together, while also expanding the breadth of the data. As a result, this technique opens up the possibility of exploring gene set evolution in non-model organisms.

TN-seq has been broadly applied in bacterial genetics where it revolutionized functional systems genomics [24,28,29]. While only used lightly in eukaryotes, it was pioneered in fungal genetics in the model fission yeast *Schizosaccharomyces pombe* using the Hermes transposon [30]. The Hermes transposon generates single insertions in vivo in *S. pombe* with relatively little insertional bias [30,31]. In addition, the extremely high insertion density reduces the likelihood of false positives or false negatives resulting from analyzing only one mutant per gene.

Subsequent studies in fission yeast have used TN-seq to identify factors involved in heterochromatin formation [32] and to study the fitness landscape of the genome [33]. More recently, TN-seq systems have also been developed in two other model yeasts, *Saccharomyces cerevisiae* [34,35] and *Candida albicans* [36], as well as a few less commonly studied fungi [37–39].

Here we develop and apply an assay using TN-seq to comprehensively identify the relative contribution of the majority of genes in the genome to sexual reproduction in the model fission yeast *S. pombe*. In *S. pombe*, sex begins with the fusion of two haploid cells of opposite mating type that then undergo karyogamy (nuclear fusion). The resulting diploid then proceeds through meiosis to produce four haploid spores within a sac (ascus), analogous to the production of sperm through male meiosis in animals. When the ascus breaks down and these dormant spores encounter appropriate nutritional signals, they germinate to regenerate the original haploid yeast state. As a result, sexual reproduction requires genes involved in pheromone sensing, cell fusion, nuclear fusion, meiosis, sporulation, and spore germination.

Previous experiments have focused either on the ability to produce spores that stain appropriately with a spore dye [40], on cytological analysis of chromosome segregation and sporulation rates [17], or on mating ability [41]. In addition, other groups have explored transcripts associated with various stages, such as meiotically upregulated genes [16,42] or genes expressed during spore germination [43]. Our work is the first to directly assess the contributions of the entire genome to the whole process of sexual reproduction in *S. pombe*. As a result, we draw on the body of existing research on sex in *S. pombe* [17,40,41] to assess the validity of our TN-seq-based approach, but we also identify additional genes involved in sexual reproduction that were not identified by previous approaches. For example, we identified a new class of sexual reproduction genes whose mutants exhibit a competitive growth disadvantage during spore germination. In addition, we identified a mutant that failed to grow under conditions of low cell density, a condition likely to be frequently encountered by spores in nature and often employed in studies of reproduction. This work provides the basis for rapid exploration of the genes involved in sexual reproduction in numerous fungi, including other members of the *Schizosaccharomyces* genus. Further, our assay identified many conserved genes that can be further studied in multicellular eukaryotes, in addition to *S. pombe*.

## Results

To determine the set of genes that contribute to sexual reproduction in *S. pombe*, we developed an assay to test the ability of cells to successfully mate, undergo meiosis, and produce viable spores. We generated a transposon-mutant library in a strain capable of mating type switching ($h^{90}$) using a previously described method [30] with some minor modifications (see Methods). We then plated this library at low density onto malt extract agar (MEA) plates (Fig 1C). On MEA plates, *S. pombe* cells can undergo a small number of divisions, allowing individual cells to produce microcolonies that are clonal except for the mating type locus. Once local nutrients have been exhausted, cells within these microcolonies mate to produce diploids that are homozygous for a single transposon insertion. The diploids then quickly undergo meiosis and sporulation.

Our mutant library was similar to a previously published transposon insertion library, with enrichment of inserts in annotated non-essential genes and depletion in annotated essential genes (S1 Fig) [30]. Part of our initial mutant library was then separated into aliquots for cryopreservation, while the remainder was used to prepare sequencing libraries. To begin our sexual reproduction assay, we revived and cultured one cryopreserved aliquot. Part of this culture was used to prepare a second round of pre-sex sequencing libraries, while the remainder was

diluted and spread sparsely onto MEA plates. We isolated spores from the MEA plates and allowed them to germinate in liquid culture for either a "short" (single 1:100 dilution of spores into rich medium and grown for 24 hours) or "long" (1:1000 dilution of spores into rich medium grown for 24 hours and diluted 1:100 into rich medium and grown for another 24 hours) outgrowth period. Sampling two timepoints prior to sex and two timepoints after sex allowed us to assess when changes in insert frequencies were likely to be affected by growth rate.

To identify and quantify individual transposon insertion sites at each timepoint, we amplified transposon-associated DNA via PCR and carried out Illumina sequencing with some modifications from previous protocols (see Methods) [30,33]. Importantly, our sequencing library preparation added unique barcodes to each genome fragment, as previously employed in *S. pombe* [33]. These barcodes allow us to correct for bias in PCR amplification and directly assay the frequency of a given transposon insertion in our mutant library [33].

We mapped only sequencing reads containing the PCR-amplified Hermes transposon fragments and trimmed the Hermes sequence from the reads prior to mapping. To avoid artifacts derived from repeat regions, we only used reads that mapped uniquely to the *S. pombe* genome (approximately 95% of the genome). In sum, we identified 859,876 unique transposon insert locations in our original mutant library, of which we recovered 655,561 after sampling, freezing, and regrowing. After sexual reproduction, we recovered 475,815 unique insert locations (72.6% of those in the plated culture) after a short outgrowth and 458,221 (69.9%) after a longer outgrowth. Insert sites present in the mutant library prior to inducing sexual reproduction can be absent from the post-sexual reproduction sets for multiple reasons. First, they may have a bona fide defect in sexual reproduction. Alternately, inserts into essential genes whose protein product has a long turnover may not be immediately eliminated from the population, but may have been lost over the course of our sexual reproduction assay. In addition, mutants that failed to form microcolonies on MEA would also be expected to be depleted in our post-sexual reproduction data sets. Finally, some inserts present at low frequency in the initial set may have failed to be plated by chance and thus were not present at later time points for non-biological reasons.

To reduce noise from poor sampling of low frequency inserts, we filtered out all sites that were represented by eight or less total unique ligation products in our sample taken from the mutant library prior to inducing sexual reproduction, as determined by reads at the same sites with unique barcodes added during the library preparation. This reduced our total number of unique insert sites to 364,549 prior to inducing sexual reproduction, 326,269 (89.4%) in the short outgrowth post-sexual reproduction data set, and 318,729 sites in the longer outgrowth post-sexual reproduction set (87.4%). This filtering likely reduced our ability to detect true positives but also reduced the likelihood to detect false positives. The remaining mutants still represent a high insert density across the genome with one insert every 38 bases on average and did not result in any gross changes in insert distribution (S2 Fig).

## TN-seq identified numerous sexual reproduction candidate genes

To assess the role of a gene in sex, we measured the frequency of individual inserts in that gene within the mutant library prior to inducing sexual reproduction (pre-sex), as well as in the samples taken from germinated spores (post-sex). Inserts within many known meiotic genes displayed a clear reduction in frequency after sex, as seen for *rec12* [44,45], the ortholog of the well-studied *SPO11* endonuclease required for meiotic recombination (Fig 2A) [46]. For a more comprehensive analysis, we built a distribution of log-corrected fold changes (log(insert frequency after sex/insert frequency before sex)) for insert sites across each gene. Because a log

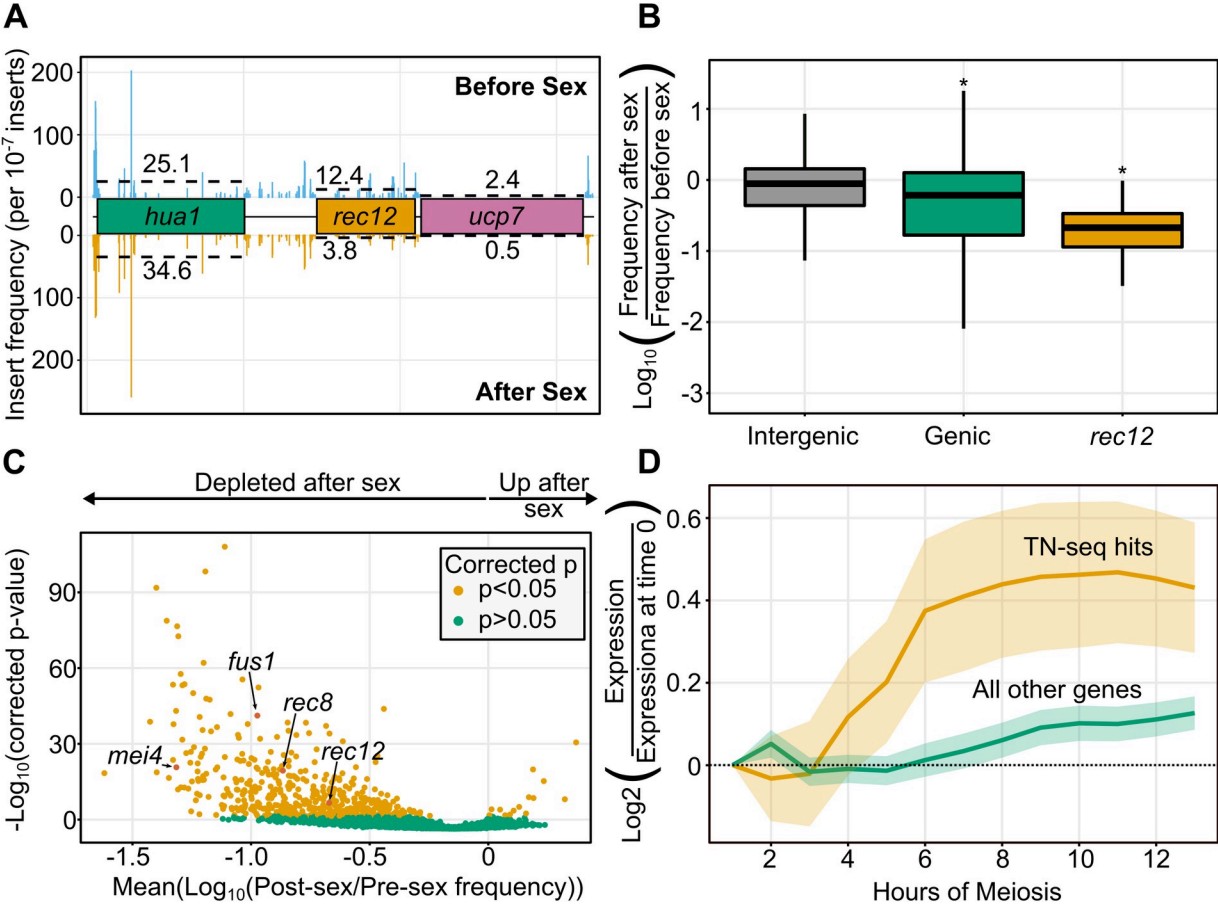

**Fig 2. Sexual reproduction TN-seq identifies bona-fide sex genes as well as numerous candidate genes.** A) Plot of transposon insert frequency across a stretch of the genome centered on the *rec12* (*SPO11*) gene. Frequencies before sex are indicated with bars above the gene cartoons and frequencies after sex are indicated with bars below the gene cartoons. The dotted line indicates mean frequency within the library of insert sites. Genes are color coded either as dispensable (green), important for sexual reproduction (orange), or essential (pink). Insert frequency is scaled to frequency per $10^{-7}$ inserts. B) Boxplot displaying distribution of $\log_{10}$-adjusted fold changes in insert density after sex (ie. Frequency after sex/frequency before sex). Boxplots show first quartile, median, third quartile. The whiskers show the range to a maximum of 1.5 times the interquartile range above and below the first and third quartile, respectively. Outlier data points (outside the whiskers) are not displayed. This results in 5,716 of 235,578 intergenic sites, 234 of 128,971 from coding regions, and 2 of 36 from *rec12* not being displayed, although those data were considered in the statistical analyses. Inserts in intergenic regions are indicated in grey, inserts into genic regions (including introns) are shown in green, and inserts into the known meiotic gene *rec12* are shown in orange. C) Volcano plot of 3418 genes with 5 or more insert sites (out of 5118 pombe genes) displaying the mean $\log_{10}$ (post-sex/pre-sex) value on the x-axis and the -$\log_{10}$ (Bonferroni corrected p-value) on the y-axis. Individual genes are shaded orange if the distribution of inserts is statistically different from the distribution of inserts into noncoding regions (p<0.05 via Mann-Whitney U test after Bonferroni correction). Genes that are not statistically different are shaded green. Four genes with known sexual reproduction defects are highlighted. D) Plot of average gene expression across meiosis (from [49]) for genes identified as hits with defects in sexual reproduction (orange) or not (green).

(0/x) is undefined, we set the minimum detected value for any given insert site to one read, so that sites with no detected reads in a subset of conditions were instead assumed to have one read in those conditions. By building a distribution of insert frequencies at each site within each gene, we were able to compare the distribution of log-corrected fold changes across a gene to that of a putatively neutral distribution using a Mann-Whitney U test. For a "neutral" control set, we analyzed insert frequencies in intergenic regions of the genome, which had a mean log-adjusted ratio slightly below zero (i.e., very little change in insert frequency between the pre- and post-sex samples) (Fig 2B). Sexual reproduction genes were defined as those

which had a statistically different distribution of log-corrected fold changes from that of the intergenic region after Bonferroni multiple testing correction (Fig 2C).

This approach is conservative and likely underestimates the total number of sexual reproduction genes. In addition, we did identify two categories of likely errors. First, a subset of genes had a very low number of unique insert sites and thus were difficult to score via this approach. These genes typically had either very short coding sequences or were annotated essential genes that had retained very low insert numbers at our early time points. To address this, we filtered to remove all genes which had fewer than five unique insert locations. Of the 5,118 genes annotated in *S. pombe*, 4,435 had at least one unique insert site in our library. 3,418 had at least five unique insert sites, meaning that we were able to assess function in sexual reproduction for approximately two-thirds of the *S. pombe* genome in a single pool. Of those 3,418 genes, 532 were inferred to have a role in sexual reproduction via our assay in at least one of the two outgrowth conditions, while 15 were predicted to repress sexual reproduction. After this filtering, there were still 10 remaining genes that were annotated as essential. Previous assays using TN-seq have revealed that some essential genes tolerate insertions over relatively small portions of the coding sequence and thus TN-seq can identify non-essential regions of essential genes or generate alleles with decreased but not eliminated function [32]. These remaining hits could be false positives resulting from incomplete loss of an essential gene or they could reveal sexual reproduction roles for essential genes. Alternately, they could reveal incorrect annotations for essentiality. In fact, at least one of these "essential" genes (*SPAC12B10.02c*, described below) was not essential in our hands.

We identified another caveat by examining the genes our assay identified as repressors of sex. Genes whose mutants have either substantial vegetative growth advantages or disadvantages may appear to have advantages or disadvantages during sex simply because of their vegetative growth rate. We identified 15 genes that appeared to function as suppressors of sexual reproduction (i.e., inserts statistically increased in frequency over meiosis). However, our assay allowed us to measure the change in insert frequency at two vegetative growth steps, one before sex and one after spore germination. Most of the candidate repressor genes (12/15) increased in frequency at both vegetative growth steps (S3 Fig). Thus, most of these genes may not be suppressors of meiosis but instead limit vegetative growth rate. As a result, we focused on genes required for meiosis and with relatively little impact on vegetative growth. We applied the same statistical test to growth rate before sex and to growth rate after spore germination and found only 21 of our 532 candidate sexual reproduction genes appeared to be growth impaired prior to our sexual reproduction assay and only 46 were statistically growth impaired after our sexual reproduction assay (S1 Table). Importantly, only three of these genes were impaired at both stages (*Pvg1*, *Meu10*, and *Put2*), giving us confidence that the vast majority of the phenotypes we observed were the result of defects in sexual reproduction rather than growth rate.

Of our 532 candidate sexual reproduction genes, 17 are *Schizosaccharomyces* specific, 509 are conserved in fungi, and 374 are conserved in vertebrates (S1 Table, S4A Fig). 40 of our hits lack a *S. cerevisiae* ortholog but are conserved in vertebrates. 45 hits lack an assigned name in *S. pombe*, and of those, 9 lack an ortholog in the budding yeast *S. cerevisiae* but are conserved in vertebrates (S1 Table) [47].

Our data are consistent with previous studies of sexual reproduction in *S. pombe*. 296 of our hits had previously annotated phenotypes in sexual reproduction. Further, transposon insertions into genes annotated in PomBase with decreased spore germination frequency in deletion mutants (FYPO:0000581) are significantly underrepresented in the post-meiosis dataset ($p = 5.6^*10^{-9}$ Mann-Whitney U test) [48]. Similarly, inserts into genes annotated with abnormal meiotic chromosome segregation (FYPO:0000151) are also underrepresented

($p = 6.3^*10^{-4}$, Mann-Whitney U test). Finally, our set of genes with a function in sexual reproduction are generally more highly expressed during meiosis (Fig 2D) [49].

However, there are obvious caveats to our data as well. Numerous genes in our set of hits are required for biosynthesis of essential metabolites, such as *arg1*, *arg4*, *arg11*, *cys11*, *lys1*, and *lys2*, among others. These hits may reflect a deficiency of those molecules in the MEA used for our TN-seq assay rather than a specific requirement for these genes to undergo sex and meiosis. However, multiple genes annotated with auxotrophy phenotypes also have existing phenotypes in sexual reproduction. For example, *SPAC27E2.01* and *Hmt2* are auxotrophic for arginine [50] and cysteine [51], respectively. In addition to auxotrophy, *SPAC27E2.01* mutants have an annotated defect in sporulation and *Hmt2* mutants have an annotated defect in meiotic chromosome segregation [17]. Likewise, many genes we identified as important for sex also have annotated roles in metabolic or biosynthetic processes (159 genes, including auxotrophs) (S4B Fig). As above, many of these genes (65 genes) have existing sexual reproduction annotations. For example, ribosomal subunits, collectively required for one of the most core functions in the cell, have annotated defects in meiotic chromosome segregation (*Rpl26* and *Rps1702*) [17] and in mating (*Rps101* and *Rps1801*) [41]. It is quite likely that many genes with existing known functions have pleiotropic roles and play multiple important roles throughout the life of a cell, including in sexual reproduction.

20 of the genes with known sexual reproduction functions are involved in autophagy (17 *atg* genes, *isp6*, *ctl1*, and *fsc1*), a process known to be required for sexual reproduction in *S. pombe* [52,53]. Of the remaining 142 genes without an annotated metabolic role or sexual reproduction phenotype, 98 are conserved in vertebrates (S1 Table).

## Candidate genes identified by TN-seq display sexual reproduction defects

To validate our assay, we focused on four genes. Three of these genes displayed sexual reproduction defects in our TN-seq assay but lacked a relevant sexual reproduction annotation: SPAC3G6.03c (named herein *ifs1* (important for sex)), *plb1*, and *alg9*. We chose *ifs1*, *plb1*, and *alg9* because they all had statistically significant and very large magnitude substantial defects in our TN-seq assay (average post-sex frequency dropped more than an order of magnitude) but were not previously described as defective in sexual reproduction. The fourth gene, *atg11*, was previously annotated with a meiotic function but was a borderline hit in our assay that was statistically different from neutral before multiple test correction, but not after (Fig 3A). *ifs1* encodes a maf-like protein that has not been explored in *S. pombe*, but homologs in bacteria act to inhibit septation [54]. *plb1* encodes a phospholipase B enzyme, which is involved in lipid metabolism. Mutants of *plb1* exhibit sensitivity to osmotic stress as well as loss of nutrient repression of mating at 20˚C, resulting in an increase in mating efficiency rather than the poor fitness observed in our assay [55]. *alg9* encodes an alpha-1,2-mannosyltransferase that has no obvious link to meiosis aside from one report of a weak physical association with *moc3* and *moc1*/*sds23*, which are important for sexual differentiation [56].

In order to test the hypothesis that these genes are important for sexual reproduction, we began by individually deleting each gene in a common strain background. Mutants of all four genes successfully mated and formed asci on MEA, although *plb1Δ* mutants produced asci that were unusually flocculant (i.e., clumpy) (S5 Fig). We quantified the frequency of typical four-spore tetrads for each mutant (Fig 3B). Both *plb1Δ* and *atg11Δ* were significantly different from wild type (Fisher's exact test, wild type, n = 287; *alg9Δ*, n = 337, p = 1.0; *plb1Δ*, n = 271, p = 0.031; *atg11Δ*, n = 491, p = 0.0004), although in opposite directions, with *plb1Δ* mutants producing a slightly higher percentage of four-spore tetrads and *atg11Δ* mutants producing slightly less.

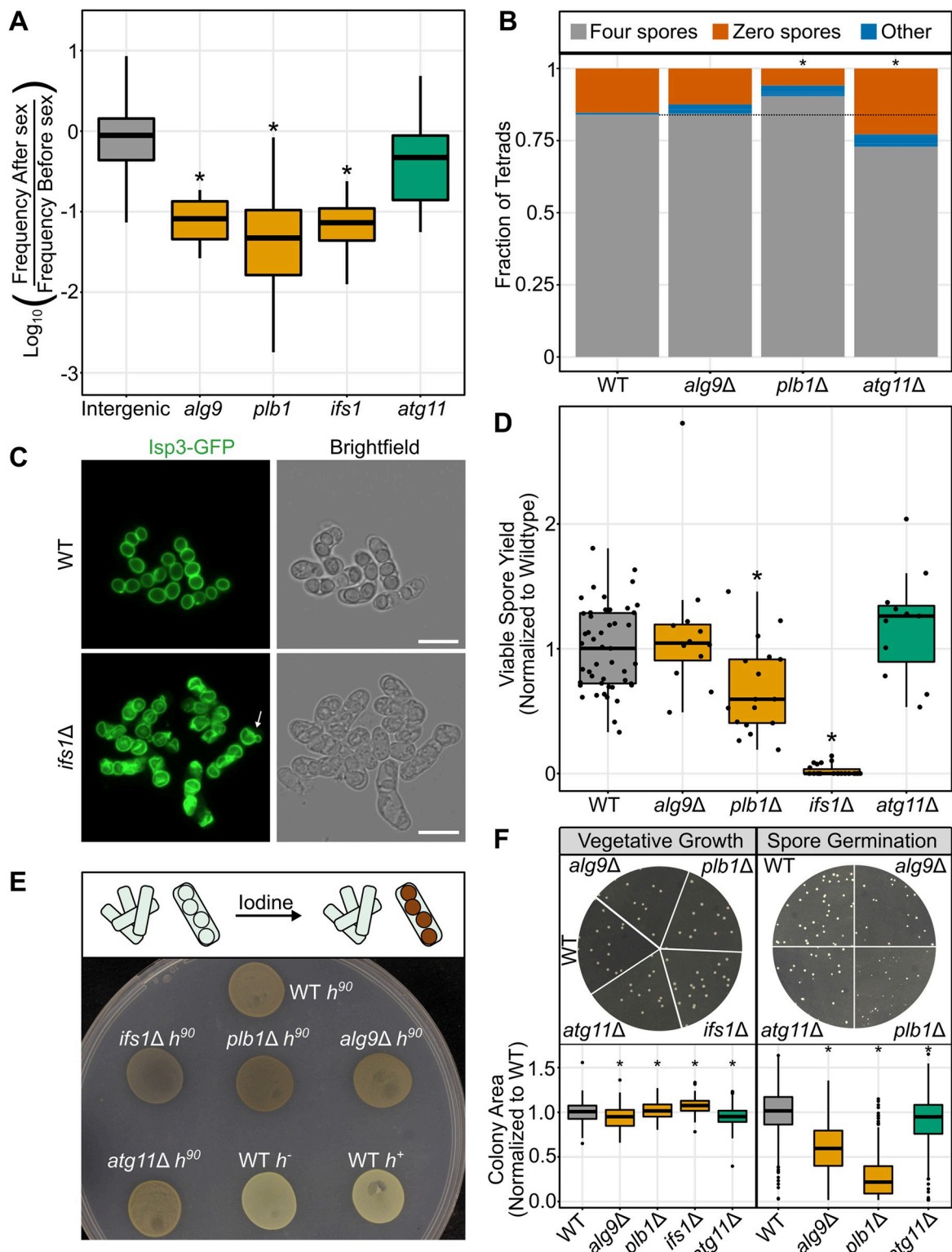

**Fig 3. TN-seq identified candidates have defects in spore germination and spore health.** A) Boxplot displaying distribution of log10-adjusted fold changes in insert density after sex (ie. frequency after sex/frequency before sex). Boxplots show first quartile, median, third quartile. The whiskers show the range to a maximum of 1.5 times the interquartile range above and below the first and third quartile, respectively. Outlier data points (outside the whiskers) are not displayed. This results in 5,716 of 235,578 intergenic sites, 2 of 16 sites from *alg9*, 0 of 166 sites from *plb1*, 2 of 38 sites from *ifs1* and 0 of 25 from *atg11* not being displayed although those data were considered

in the statistical analyses. Inserts in intergenic regions are indicated in grey, genes where inserts were significantly depleted after sexual reproduction in the TN-seq data are shown in orange (*alg9*, *plb1*, *ifs1*) and genes where inserts were not significantly depleted after sexual reproduction are shown in green (*atg11*). B) Scoring of acsi produced by the indicated mutants and wild type after 2 days at 25˚C on MEA plates. Asci were scored as having either four spores, no visible spores, or any other category (1, 2, 3, or more than 4). *ifs1*Δ mutants were not scored because the spores were atypical in appearance and difficult to accurately identify. Fisher's exact test, WT, n = 287; *alg9*Δ, n = 337, p = 1.0; *plb1*Δ, n = 271, p = 0.031; *atg11*Δ, n = 491, p = 0.0004. C) Isp3-GFP was visualized in wild type and *ifs1*Δ mutants on an AXIO Observer.Z1 (Zeiss) wide-field microscope with a 40x C-Apochromat (1.2 NA) water-immersion objective after incubation for 2 days at 25˚C on an MEA plate. Scale bars indicate 10 microns. Arrow indicates "snowman" spore. D) Viable spore yield assay showing on the y-axis the number of spores produced per yeast cell plated, normalized to the mean value for wild type. Cells were incubated on MEA plates in a dense growth spot for 3 days at 25˚C prior to spore isolation. All mutants were assayed in a set of at least 5 biological replicates alongside at least 5 wild type replicates. Points display results from a single replicate, normalized to the mean from the corresponding wild type controls. The boxplots summarize the underlying points and show first quartile, median, third quartile while the whiskers show the range of the data to a maximum of 1.5 times the interquartile range below and above the first and third quartile, respectively. Points outside the whiskers can be considered outliers. Mann-Whitney U Test, *alg9*Δ, p = 0.61; *plb1*Δ, p = 0.0028; *ifs1*Δ, p = $1.18^{*}\ 10^{-11}$; *atg11*Δ, p = 0.21. E) Iodine staining of $h^{90}$ mutants after 3 days on SPAS medium at 25˚C. Iodine should stain spores brown, while leaving unsporulated cells unstained (top). Wild type $h^{90}$ is shown as an iodine-staining positive control and wild type $h^{-}$ and $h^{+}$ strains are shown as non-staining negative controls. F) Plates illustrating colony size for wild type and mutant strains after 3 days of growth at 32˚C on YEA +SUP plates. Vegetative growth was plated from a serial dilution of an overnight culture in YEL+SUP at 32˚C. Spore growth was plated from a serial dilution of glusulase-treated spores that were originally generated after 3 days on MEA at 25˚C. All spores were generated, plated, and imaged at the same time. The bottom plot displays quantification of colony sizes using Fiji to perform thresholding and particle analysis. The boxplots summarize measurements from 4–5 separate plate images for each genotype and show first quartile, median, third quartile while the whiskers show the range of the data to a maximum of 1.5 times the interquartile range below and above the first and third quartile, respectively. Points outside the whiskers can be considered outliers. All four mutants produced yeast colonies that were on average statistically significantly different in size from wild type (Mann-Whitney U test, wild type, n = 181; *alg9*Δ, n = 183, p = $4.3^{*}\ 10^{-6}$; *plb1*Δ, n = 213, p = 0.033; *ifs1*Δ, n = 210, p = $2.8^{*}\ 10^{-12}$; *atg11*Δ, n = 164 p = 0.00034) All three mutants tested produced spore colonies that were statistically significantly smaller than wild type (Mann-Whitney U test, wild type, n = 467; *alg9*Δ, n = 213, p<2.2 $^{*}\ 10^{-16}$; *plb1*Δ, n = 388, p = $<2.2^{*}\ 10^{-16}$; *atg11*Δ, n = 286, p = $3.3^{*}\ 10^{-6}$).

*Ifs1*Δ mutant asci were difficult to score via brightfield imaging because spores did not appear to properly individualize, although vaguely spore-like shapes were sometimes visible within asci (S5 Fig). To better visualize *ifs1*Δ mutant spores, we generated a deletion of *ifs1* in an *isp3*-GFP strain. Isp3 coats the exterior of wild type spores (Fig 3C, top) [57]. Imaging Isp3-GFP in asci generated on MEA medium revealed that *ifs1*Δ mutants made wrinkled, irregular spores that frequently had blebs on the exterior, reminiscent of previously described "snowman" spores (Fig 3C, bottom) [58]. This phenotype was less severe for asci generated on SPA medium, where spores were still frequently snowman-shaped, but otherwise had less dramatic structural defects (S6 Fig). Because imaging asci generated on MEA required scraping cells off of plates to image, while spores could be imaged in place on SPA, it is possible that *ifs1*Δ spores are more fragile than wild type spores and the more severe phenotype on MEA is simply the result of increased manipulation for imaging.

We next used a viable spore yield (VSY) assay to assess fertility. Briefly, viable spore yield measures the total number of viable spores produced per cell plated on the original mating plates [59]. Two mutants (*ifs1*Δ and *plb1*Δ) had statistically decreased viable spore production via this assay after three days incubating on MEA mating plates in dense cell patches (Mann-Whitney U Test, *alg9*Δ, p = 0.61; *plb1*Δ, p = 0.0028; *ifs1*Δ, p = $1.18^{*}\ 10^{-11}$; *atg11*Δ, p = 0.21) (Fig 3D). The phenotype of *ifs1*Δ was very strong. Spores very rarely germinated to form colonies, which was unsurprising given the extreme spore formation defects we observed. We retested two *ifs1*Δ spore colonies via viable spore yield to test the hypothesis that a rare genetic suppressor had arisen in these surviving spores that rescued the *ifs1*Δ mutant phenotype. However, these survivors retained the mutant phenotype, suggesting instead that the function of *ifs1*, while extremely important, is not absolutely required for spore germination (S7 Fig).

Previous studies of *S. pombe* have utilized the fact that sexual spores will stain brown when exposed to iodine while vegetative cells will not to rapidly screen for mutants with defects in sexual reproduction (Fig 3E, top) [13,40,60,61]. This phenotype is visible without magnification in patches on mating media so it is simple to screen large numbers of mutants. Because

we knew *ifs1Δ* mutants produced very few viable spores, we tested whether they or any of the rest of our mutants produced patches that stained with iodine. All four of our tested mutants stained with iodine on mating media, indicating that even though *ifs1Δ* spores almost never germinate and appear atypical via microscopy, they still stain with iodine (Fig 3E, bottom). This may help explain why some of our TN-seq hits were missed in past screens that used iodine to look for sporulation defects in *S. pombe*.

### *plb1Δ* mutants exhibit a gamete health defect

While the defect in the *ifs1Δ* mutant was nearly complete, *plb1Δ* mutants had a much more subtle defect via the viable spore yield assay and both *atg11Δ* and *alg9Δ* mutants appeared to have no defect at all (Fig 3D). Because our TN-seq data led us to expect a strong defect in viable spore yield for *plb1Δ* and *alg9Δ*, we assayed viable spore yield for these mutants under other conditions but did not reproduce the defect in reproductive fitness for either mutant detected in our TN-Seq assay (S8A-S8D Fig). In fact, reproducing the conditions originally used in the TN-seq assay for the viable spore yield assay generated a small *increase* in viable spore yield for *alg9Δ* (Mann-Whitney U Test, *alg9Δ*, p = 0.0022; *plb1Δ*, p = 0.24; S8D Fig). Combining all the data generated for these mutants in this study across multiple conditions resulted in no distinguishable defect in viable spore yield relative to wild type (Mann-Whitney U Test, *alg9Δ*, p = 0.0013 with higher mean than wild type; *plb1Δ*, p = 0.73) (S8E Fig).

However, we noticed that colonies produced by germinating *plb1Δ* mutant spores were variable in size and were often substantially smaller than those produced by wild type spores (Fig 3F, top right). Colonies derived from *alg9Δ* mutant spores similarly produced visually smaller colonies than wild type spores. To assess this quantitatively, we imaged colonies generated from spores for wild type, *alg9Δ*, *plb1Δ*, and *atg11Δ* mutants and measured colony size computationally. We did not test *ifs1Δ* mutant spores because their extremely low germination rate made this experiment impractical. As expected, *alg9Δ* and *plb1Δ* mutants produced smaller colonies, but surprisingly, *atg11Δ* mutant spores also had a relatively subtle growth defect as well (Mann-Whitney U test, p = $3.3 * 10^{-6}$) (Fig 3F, bottom right). As a control, we performed the same assay by plating vegetatively growing cells and measuring colony sizes. All four mutants tested were significantly different from wild type, but with substantially smaller magnitudes of change (Fig 3F, bottom left). Notably, *ifs1Δ* and *plb1Δ* mutants made slightly larger colonies than the wild type on average.

As a result, we predicted that when grown in competition with wild type spores, *plb1Δ*, *alg9Δ*, and *atg11Δ* mutant spores would have a far more dramatic fitness disadvantage during spore germination resulting from either delayed germination or slow growth following spore germination. To test this hypothesis, we performed a competitive growth assay (Fig 4A). We mixed approximately equal ratios of wild type and mutant spores in rich liquid medium to allow them to germinate and grow. We took an initial sample to verify that we had properly mixed our spores. Then we allowed growth for 24 hours and plated again. We assessed the frequency of the two genotypes by replica plating colonies to medium containing G418. Wild type cells die on this medium, but each of our mutants survive because we replaced the target coding sequence with the *kanMX4* gene. As a control, we also performed the same assay by competing a presumably neutral deletion of a *wtf12* pseudogene marked with the same selective marker [62,63] against the same wild type strain. This assay revealed that both *plb1Δ* and *alg9Δ* mutant spores had a substantial competitive growth defect, while *atg11Δ* mutant spores once again had a relatively subtle defect (Mann-Whitney U test, *alg9Δ*, p = $1.9 * 10^{-10}$; *plb1Δ*, p = $1.5 * 10^{-8}$; *atg11Δ*, p = 0.0483) (Fig 4B, right). We also performed the same assay starting with vegetatively growing yeast instead of sexual spores. Unlike spore germination, *plb1Δ* and

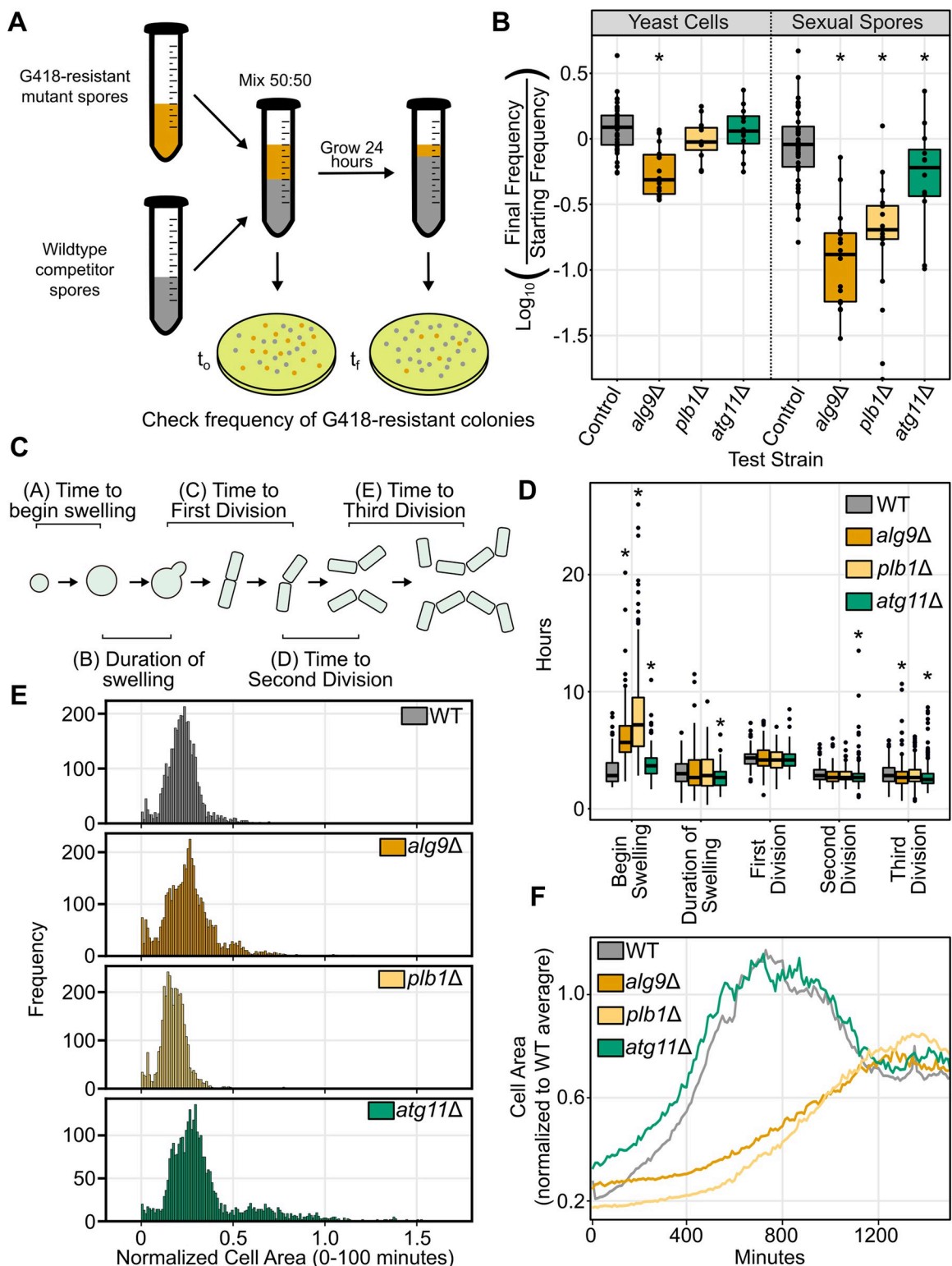

**Fig 4. Mutant spores are less competitive and germinate more slowly.** A) Schematic showing spore competition assay. Spores are mixed at a 50:50 ratio and incubated with shaking overnight at 32°C in rich medium (YEL+SUP). Samples of starting and final cultures are plated to YEA+SUP medium. Once colonies grow, they are replicated to YEA+SUP with G418 (100 mg/L) and the proportion of G418-resistant colonies is counted. B) Results of spore and yeast competition experiments. Results are a log ratio of the final frequency normalized to the initial frequency. Points indicate results of individual replicates from two experiments, each including at least 5

biological replicates. The boxplots summarize the underlying points and shows first quartile, median, third quartile while the whiskers show the range of the data to a maximum of 1.5 times the interquartile range below and above the first and third quartile, respectively. Points outside the whiskers can be considered outliers. C) *S. pombe* spores undergo several landmarks in the process of germinating. Spores are initially dormant until they begin growing isotropically (ie. swelling). This phase ends when cells begin polarized growth and elongate on one side. This cell will elongate until it eventually divides by fission to produce two daughters. Each of those daughters will go on to divide a second time and eventually a third time. We scored each of these landmarks manually from videos using Fiji. D) Videos of spore germination in a CellASIC device with flowing YEL+SUP medium at 32˚C were scored and the time between each step in spore germination was tracked for individual spores. The boxplot shows first quartile, median, and third quartile while the whiskers show the range of the data to a maximum of 1.5 times the interquartile range below and above the first and third quartile, respectively. Points outside the whiskers can be considered outliers. (For time to initiation of swelling, duration of swelling, time from polarization to first division, time from first to second division and time from second to third divisions, respectively: wild type n = 113, n = 115, n = 114, n = 133, n = 262; *plb1Δ* (n = 113, p<2.2 $^*$ 10$^{-16}$; n = 80, p = 0.764; n = 71, p = 0.3202; n = 130, p = 0.263; n = 225, p = 0.4623); *alg9Δ* (n = 59, p<2.2 $^*$ 10$^{-16}$; n = 85, p = 0.895; n = 85, p = 0.7515; n = 162, p = 0.09212; n = 281, p = 0.00078); *atg11Δ* (n = 145, p = 1.48 $^*$ 10$^{-5}$; n = 142, p = 0.0029; n = 143, p = 0.295; n = 234, p = 0.00039; n = 308, p = 1.32 $^*$ 10$^{-8}$)). All p values are for comparisons to the same wild type stage. E) Histogram of spore sizes from the first 10 frames (100 minutes) of videos of spore germination. Mutant and wild type are each derived from at least two videos each from two separate days. Spores were identified using deep learning (see methods). To account for small changes in the detection of spores via deep learning, the spore areas were normalized for each data set. To do this, the average expanded spore area was determined for the wild type strain for each data set. All spore areas were then normalized to this average area to allow comparisons between data sets taken on different days. One spore over a normalized area of twice the WT average was dropped from the *atg11Δ* plot for presentation purposes. F) Plot of average cell area over the course of videos of spore germination. Spores were identified via deep learning. Data are derived from at least two videos each from two separate days and from at least two independently prepared age-matched spore preparations. Average cell sizes stabilize once spores begin to divide and grow vegetatively as yeast cells.

*atg11Δ* mutants were not defective in competitive vegetative growth in our assay, suggesting a defect specifically during spore germination (Fig 4B, left). In contrast, *alg9Δ* mutants did exhibit a vegetative growth defect in this assay, although the magnitude was much smaller than that observed for spores and was not sufficient to explain the defect in spore growth (Mann-Whitney U test, *alg9Δ*, p = 3.3 $^*$ 10$^{-7}$; *plb1Δ*, p = 0.17; *atg11Δ*, p = 0.8723).

Defects in chromosome segregation often result in both a reduction in frequency of viable spores and production of viable but slow growing aneuploid gametes. Because spores from all three mutants showed decreases in competitive fitness, we explored the hypothesis that these mutants produced aneuploids at a higher rate than wild type. To do so, we introduced a codominant marker system for chromosome III. In this system, one parent strain contains a wild type *ade6* locus while the other strain of the opposite mating type contains an *ade6* locus that has been deleted with the *hphMX6* marker [64]. As a result, progeny can only be both Ade + and hygromycin resistant if they carry both copies of this locus, suggesting chromosome III disomy. Chromosome III is the only viable single gain of chromosome aneuploidy in *S. pombe* [65]. This assay revealed that neither *plb1Δ*, *alg9Δ*, or *atg11Δ* mutants exhibit elevated levels of disomy for chromosome III (S2 Table), suggesting that their slow growth defects are instead the result of a distinct defect in spore fitness.

### *plb1Δ* and *alg9Δ* mutant spores germinate more slowly than wild type spores

To further examine the spore fitness defects we observed, we imaged spore germination using a CellASIC microfluidics platform. Briefly, we loaded spores into a growth chamber using flow, where they were trapped and imaged over 24 to 48 hours at 32˚C as we flowed fresh medium through the chamber. Because the microfluidics chip has multiple channels, we could image spore germination for spores from wild type, *plb1Δ*, *alg9Δ*, and *atg11Δ* mutants simultaneously. We began by manually scoring several landmarks during *S. pombe* spore germination (Fig 4C). This revealed a dramatic difference in initial growth rate following spore germination (Fig 4D). Both *plb1Δ* and *alg9Δ* mutants took much longer than wild type spores to initiate swelling/isotropic growth, while *atg11Δ* also showed a more moderate delay (Mann-Whitney U test, Swelling, *plb1Δ*, p< 2.2 $^*$ 10$^{-16}$; *alg9Δ*, p< 2.2 $^*$ 10$^{-16}$; *atg11Δ*, p = 1.48 $^*$ 10$^{-5}$). After this

landmark, all three mutants were indistinguishable from wild type or even proceeded slightly faster through the landmarks (*alg9Δ*, third division, p = 0.00078; *atg11Δ*, polarizing, p = 0.0029; second division, p = 0.00039; third division, p = $1.32 * 10^{-8}$).

As a second method to confirm our assay, we used a traditional plate-based assay to investigate spore fitness for the *plb1* mutant, which had the strongest phenotype in our CellASIC assay. Briefly, we plated either wild type or *plb1Δ* mutant spores densely onto an agar plate. Immediately after the plates dried, we took a punch from the plate, placed it onto a coverslip, and began continuous imaging on a wide field scope at 32˚C for 24–48 hours. As in our microfluidic approach, we began by manually scoring several landmarks during *S. pombe* spore germination (S9A Fig). Because of noise early in the imaging, we were unable to score the initial delay prior to isotropic growth with this approach and crowding prevented scoring the third division. This approach confirmed the defect in initial growth rate following spore germination for *plb1Δ* mutants. (S9B Fig). Like before, most of the delay was prior to polarization, although with this approach, each of the two divisions tracked were also delayed, albeit more subtly than the polarization step (Mann-Whitney U test, polarization, p< $2.2 * 10^{-16}$; first division, p = 0.00072; second division, p = 0.0076).

### *plb1Δ* mutant spores are smaller than wild type spores

To extract growth parameters from the microfluidics videos, we implemented a deep learning approach to identify spores and track them throughout our imaging. This approach revealed that *plb1Δ* mutant spores were on average smaller than wild type spores (Mann Whitney U test, p< $2.2 * 10^{-16}$; wild type, n = 802; *plb1Δ*, n = 784), while *alg9Δ* mutant spores were similar in size to wild type spores (Mann Whitney U test, p = 0.1496; wild type, n = 802; *alg9Δ*, n = 994), and *atg11Δ* mutant spores were actually larger than wild type on average (Mann Whitney U test, p = $3.2 * 10^{-12}$; wild type, n = 802; *atg11Δ*, n = 720) (Fig 4E). Further, monitoring cell area throughout the videos via deep learning revealed that *plb1Δ* and *alg9Δ* mutant spores on average grew more slowly during germination than wild type spores while *atg11Δ* looked very similar to wild type (Fig 4F). Extracting growth parameters from all identified spores recapitulated what we had seen via our manual tracking, where spores first grew symmetrically in all directions before eventually establishing polarity (S10 Fig, S1 Movie). The length of our videos was limited by crowding of dividing cells and ended well before visible colonies would form to be counted in a viable spore yield assay (1–2 days versus 5 days); however, a significant portion of *plb1Δ* and *alg9Δ* mutant spores had not visibly grown by the end of the videos, despite yielding a near normal viable spore yield. Our data demonstrate that spores produced by *plb1Δ* and *alg9Δ* mutants are generally viable but exhibit substantial delays (i.e., more than 24 hours) in germination and this is the origin of the fitness defect in sexual reproduction detected in our TN-seq assay.

We were able to take the same deep learning approach to quantify spore sizes and cell growth in our plate-based data as well. This confirmed that *plb1Δ* mutants had decreased spore sizes on average (T-test, p = $2.45 * 10^{-88}$; wild type, n = 53,841; *plb1Δ*, n = 73,448) (S9C Fig). As in the microfluidics approach, *plb1Δ* mutant spores were dramatically delayed in growth relative to the wild type spores (S9D Fig). Taken together, our data suggest that in wild type cells Plb1 and Alg9 contribute to efficient spore germination and progression to cell division.

### *sdg1* is required for low density growth

Interestingly, our TN-seq analysis also identified 10 genes as significant contributors to sexual reproduction that are currently annotated as essential (Fig 5A, S1 Table) [66]. This was curious

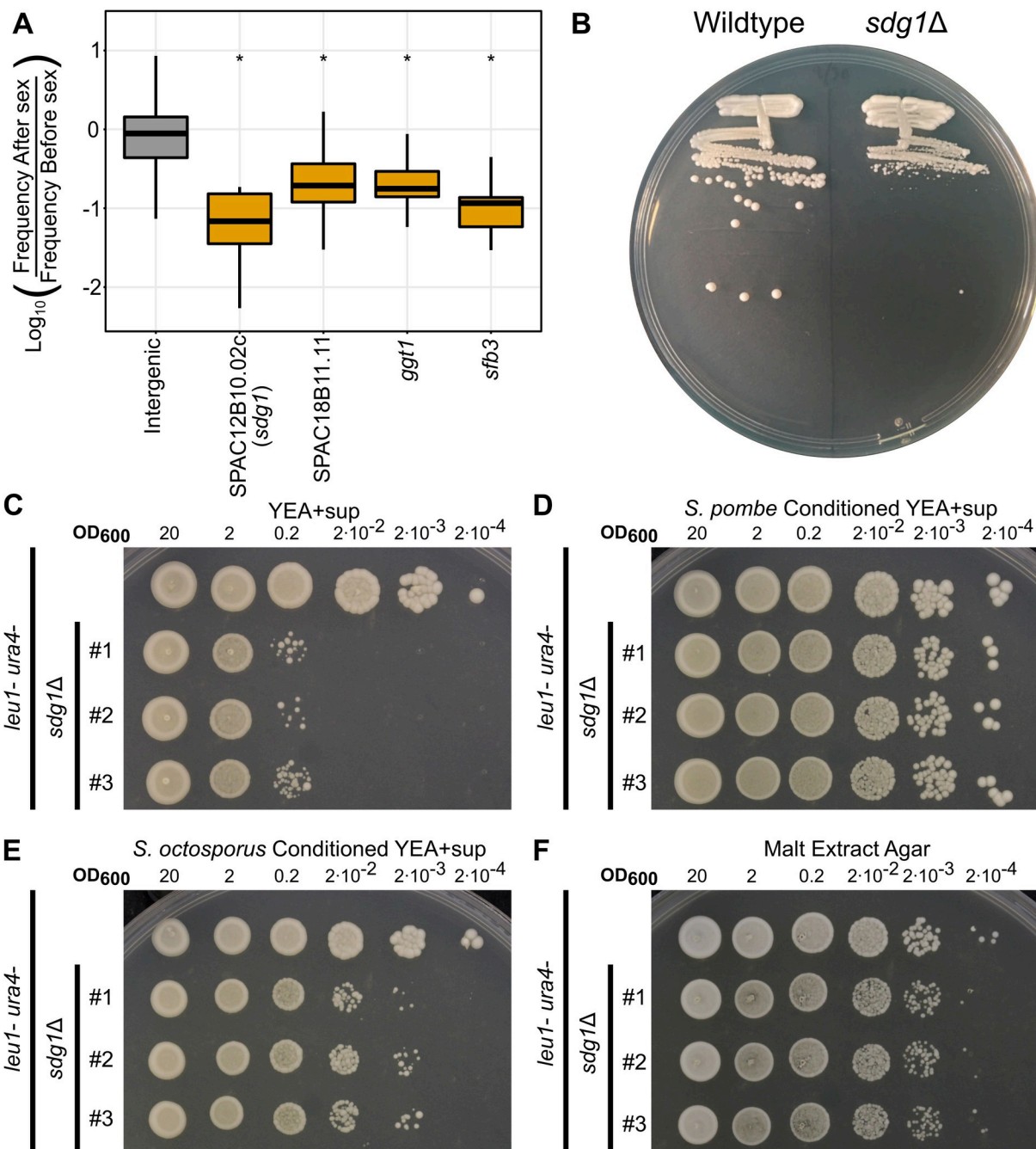

**Fig 5. *sdg1Δ* mutants have a density-dependent growth defect that can be rescued by conditioned medium.** A) Boxplot displaying distribution of $\log_{10}$-adjusted fold changes in insert density after sex (ie. frequency after sex/frequency before sex). Boxplots show first quartile, median, third quartile. The whiskers show the range to a maximum of 1.5 times the interquartile range above and below the first and third quartile, respectively. Outlier data points (outside the whiskers) are not displayed. This results in 5,716 of 235,578 intergenic sites, 0 of 24 from SPAC12B10.02c/*sdg1*, 3 of 148 from SPAC18B11.11, 2 of 22 from *ggt1*, and 1 of 14 from *sfb3* not being displayed although those data were considered in the statistical analyses. Inserts in noncoding regions are indicated in grey and inserts into candidate genes are shown in orange. B) Streaking assay showing a parent *S. pombe* strain (*ura4*-D18, *leu1*-32) and an *sdg1Δ* mutant (*sdg1Δ*::kanMX4, *ura4*-D18, *leu1*-32) struck to single colonies on a YEA+SUP plate after 4 days at 32˚C. C-F) Spot dilution assays with 5 μL spots plated. The initial leftmost spot is of $OD_{600}$ = 20 culture and each successive spot is a 10-fold dilution, so that the final spot should be $10^5$ less concentrated than the first. All four experiments were conducted on the same day with the same dilution series of parent strain (*ura4*-D18, *leu1*-32) and three independent *sdg1Δ* mutants on the same genetic background (*sdg1Δ*::kanMX4, *ura4*-D18, *leu1*-32). C) Spotted to standard yeast extract agar with supplements (YEA+SUP) and incubated for 4 days at 32˚C. D) Spotted to conditioned yeast extract agar + supplements (see methods) made from parent strain *S. pombe* (*ura4*-D18, *leu1*-32) and incubated for 4 days at 32˚C. E) Spotted to conditioned yeast extract agar made from wild type *S. octosporus* and incubated for 4 days at 32˚C. F) Spotted to malt extract agar and incubated at 25˚C for 4 days.

because our assay should only be able to query non-essential genes in which transposon insertions are present in our pre-sex sample. While some essential genes are capable of tolerating insertions over subsets of their coding sequence [32,34], we also explored the hypothesis that some of these genes were incorrectly annotated as essential. We attempted to delete two of these genes, *ggt1* and *SPAC12B10.02c*, but only successfully obtained mutants of *SPAC12B10.02c*. The *SPAC12B10.02cΔ* mutants demonstrated an unexpected "unstreakable" phenotype where cells grew at the start of a streak where the initial cell density was high, but they failed to robustly grow to a visible colony from the portions of a streak where single colonies should be present (Fig 5B). We confirmed this phenotype more formally via a spot dilution assay, where cells grew similarly to wild type at higher cell densities and failed to grow as the initial concentration of cells plated became more dilute (Fig 5C). Because this gene appears to be required for individual cells to survive and grow without other cells nearby, we named this gene *sdg1* for Social Distancing Gene. The ability to grow at low density is likely an important trait for yeast both in nature and is also important to understand in the context of laboratory analysis of sexual reproduction. Many basic assays for studying sex, such as tetrad dissection or even plating colony forming units for a viable spore assay, require cells to be plated at low densities and assume that a strain will behave similarly, independent of plating density. *sdg1Δ* mutants clearly violate these assumptions.

Very little is currently known about *sdg1* in *S. pombe*. PomBase describes it as an ortholog of *PHO86* in *S. cerevisiae*, which encodes an ER resident protein required for protein packaging into COPII vesicles [66]. However, *PHO86* and *sdg1* are only very distantly related, if at all. Three progressive rounds of searches using JACKHMMER [67], a hidden Markov Model based protein homology tool, starting from either *sdg1* in *S. pombe* or *PHO86* in *S. cerevisiae* fail to identify the other protein. JACKHMMER searches starting from *PHO86* do eventually identify other fungal proteins with the same N-acyltransferase domain as *sdg1*, including a protein from *Schizosaccharomyces cryophilus* [68] that is an ortholog of *S. pombe naa50*, rather than *sdg1*.

The density-dependent growth phenotype of *sdg1Δ* mutants was consistent with a quorum sensing effect where cell growth is dependent on factors secreted by neighboring cells. To test this hypothesis, we generated conditioned YEA+SUP (yeast extract agar with supplements) medium by replacing half of the water typically in the plates with filtered conditioned YEL +SUP (yeast extract liquid with supplements) medium previously used to grow *S. pombe* cells to saturation. We found that conditioned medium largely rescues the low-density growth defect of *sdg1Δ* mutants (Fig 5D), while simply adding more unconditioned medium did not (S11A and S11B Fig). Medium conditioned with *sdg1Δ* mutant cells conferred a similar rescue, suggesting that Sdg1 is not itself important for production of a quorum sensing signal (S11C and S11D Fig). These experiments also demonstrates that the signal responsible for rescuing low density growth survives autoclaving.

In addition, generating plates using medium conditioned by *Schizosaccharomyces octosporus* growth confers intermediate rescue of the low-density growth defect (Figs 5E and S11E), while medium conditioned by the more distant relative *Schizosaccharomyces japonicus* provided minimal rescue (S11F Fig). However, medium conditioned by the budding yeast *Saccharomyces cerevisiae* provided nearly the same level of rescue as *S. octosporus* (S11G Fig). This suggests that the quorum sensing signal is only partially conserved between these species.

Surprisingly, the low-density growth defect of *sdg1Δ* mutants was largely suppressed on MEA plates (Fig 5F). While *sdg1Δ* colonies still grew more slowly on MEA plates than on standard YEA+SUP plates, they grew at similar plated cell densities to wild type cells, which was not true when plated on YEA+SUP. We hypothesized that *sdg1Δ* mutants had performed

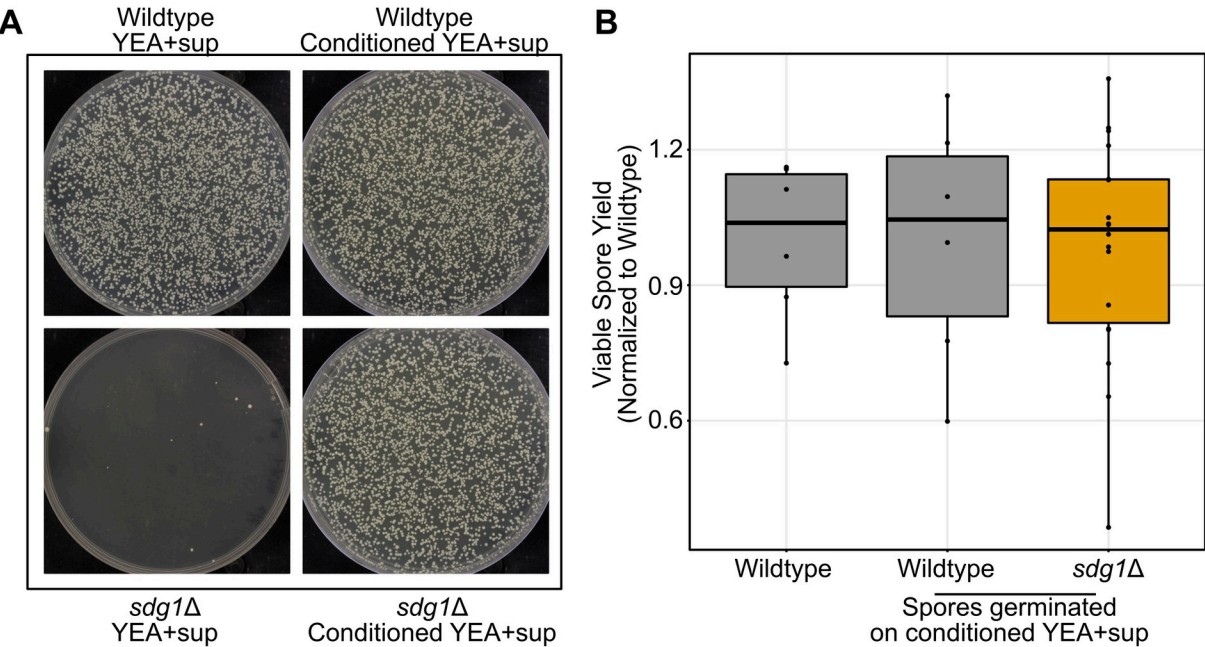

**Fig 6. *sdg1*Δ mutants do not have a sexual reproduction defect if complemented by conditioned medium.** A) Spores from wild type and *sdg1*Δ mutants were generated by incubating cells of each genotype on MEA plates for three days at 25˚C. Equal dilutions of spores from wild type and *sdg1*Δ mutants were plated to either YEA+SUP or YEA+SUP conditioned medium and incubated for 5 days at 32˚C. B) Viable spore yield assay showing on the y-axis the number of spores produced per yeast cell plated, normalized to the mean value for wild type. Points display normalized results from a single replicate. The boxplot summarizes the underlying points and show first quartile, median, third quartile while the whiskers show the range of the data to a maximum of 1.5 times the interquartile range below and above the first and third quartile, respectively. Points outside the whiskers can be considered outliers. Cells were incubated on MEA plates for three days at 25˚C in dense cell patches. Data shown for *sdg1*Δ combines three independent mutants performed on the same day. The wild type on standard medium was not different from wild type on conditioned medium (Mann-Whitney U test, p = 0.937) nor were the *sdg1*Δ mutants (Mann-Whitney U test, p = 0.923).

poorly in our TN-seq assay because we plated at low density to acquire clonal microcolonies rather than because they had a bona fide sexual reproduction defect.

However, testing viable spore yield was complicated by the fact that *sdg1*Δ mutant spores fail to grow when plated at low enough densities to count colonies accurately (Fig 6A). As a result, we employed a modified version of our assay where we plated spores to conditioned YEA+SUP medium instead of standard YEA+SUP. This did not affect the viable spore yield of wild type *S. pombe* (Fig 6B). This assay revealed that *sdg1*Δ mutants did not significantly affect viable spore yield (Mann-Whitney U test, p = 0.9225).

### Low density growth defect of *sdg1*Δ mutants is not caused by auxotrophy

A similar quorum rescuable phenotype to that of *sdg1*Δ has previously been reported in *S. pombe* (Fig 7A). This phenotype is caused by the fact that some nitrogen sources (ammonium and glutamate) inhibit nutrient uptake from the environment in *S. pombe* [69]. As a result, auxotrophs may fail to import amino acids or nucleobases and thus fail to grow even when grown in supplemented medium. However, this repression can be rescued by high cell density via oxylipins, which act as quorum molecules [70]. Because our mutants were constructed on a background auxotrophic for leucine and uracil, we hypothesized that *sdg1*Δ mutants may be a component of this previously described quorum phenotype. To test this idea, we analyzed the phenotype of *sdg1*Δ prototrophs and confirmed that mutation of *sdg1* did not itself confer aux-otrophy (S12 Fig). To our surprise, prototrophic *sdg1* mutants still exhibited a strong low

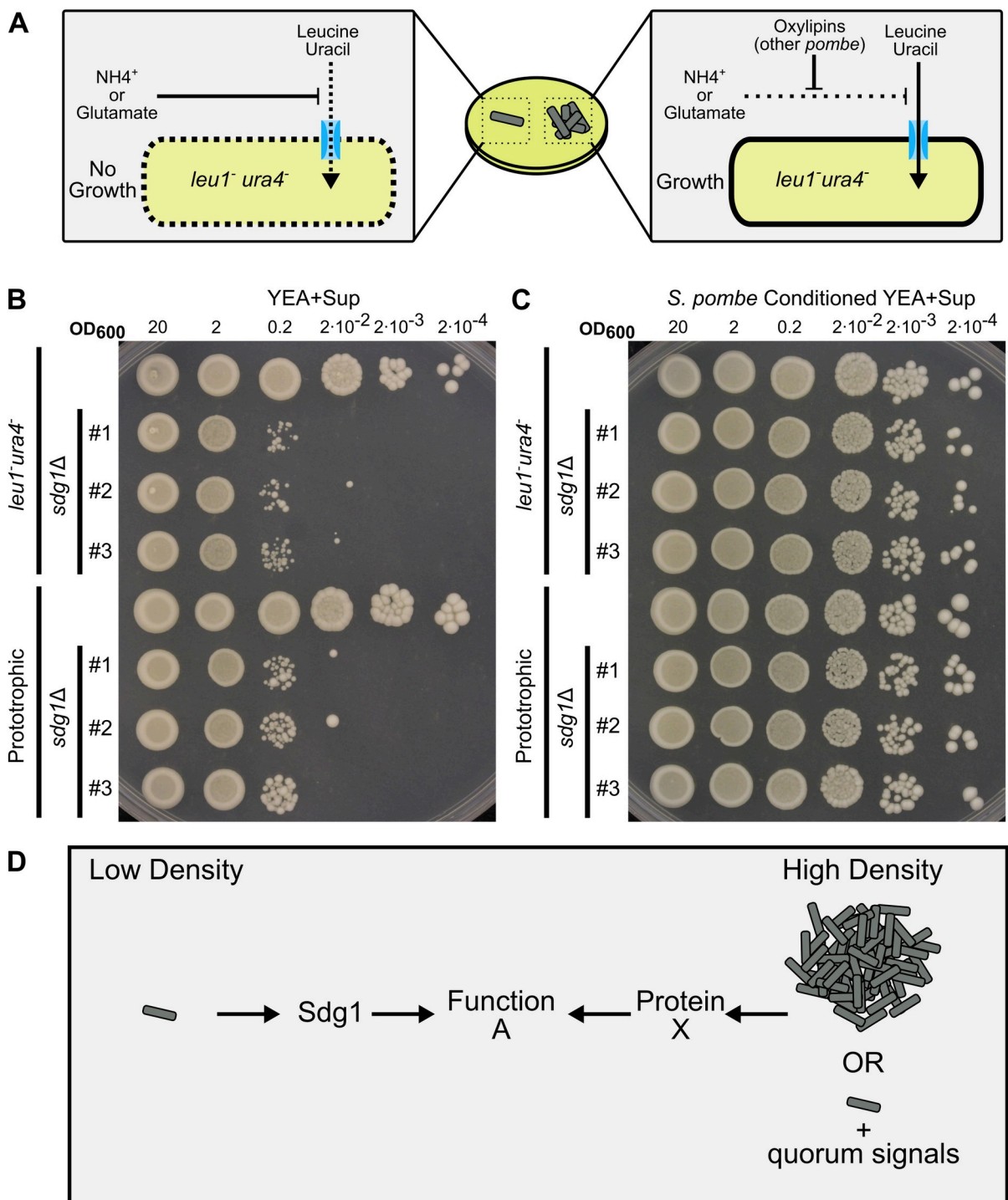

**Fig 7. The *sdg1Δ* mutant phenotype is not the result of a previously described quorum effect.** A) Diagram illustrating current model of oxylipin mediated quorum sensing in *S. pombe*. Some nutrient uptake is repressed by ammonia or glutamate in the medium. As a result, auxotrophs may fall to grow even in the presence of the appropriate amino acids (left). However, this repression can be alleviated by oxylipins that are produced as a quorum molecule (right). Consequently, many *S. pombe* auxotrophs fail to grow at low cell density. B-C) Spot dilution assays with 5 μL spots plated. The initial leftmost spot is of OD$_{600}$ = 20 culture and each successive spot is a 10-fold dilution, so that the final spot should be $10^5$ less concentrated than the first. Both experiments were conducted on the same day with the same dilution series of parent strain, three independent *sdg1Δ* mutants on the same background (*sdg1Δ::kanMX4*, *ura4*-D18, *leu1*-32), one wild type strain and three independent prototrophic *sdg1Δ* mutants (*sdg1Δ::kanMX4*). Dilution series are from different cultures from those in Fig 5B) Spotted to standard yeast extract agar (YEA+SUP) and incubated for 4 days at 32˚C. C) Spotted to conditioned yeast extract agar made from parent

strain *S. pombe* (*ura4*-D18, *leu1*-32) and incubated for 4 days at 32°C. D) Model explaining hypothesis for the *sdg1* mutant phenotype. At low density, cells use Sdg1 to accomplish some biologically necessary function A. However, at high density or at low density with exogenous quorum signals, they instead use an alternative unknown gene X. Loss of *sdg1* is thus only lethal when cells are required to grow at low density.

density growth defect (Fig 7B). As before, this low-density growth defect could be rescued by conditioned medium (Fig 7C). Notably, the existing low density growth defect of *sdg1*Δ mutants was enhanced by addition of 5 g/L ammonium chloride in auxotrophs, but not in prototrophs (S13 Fig). Together, these data suggest that the quorum-regulated phenotype we have identified here is likely distinct from the previously described oxylipin pathway [70], although it does not confirm or rule out oxylipins as the quorum molecule driving this behavior.

### Shiny app allows navigation of sexual reproduction TN-seq data

As a resource for the *S. pombe* community, we have developed an interactive Shiny web app to navigate our sexual reproduction TN-seq data (S14 Fig). This is available at (https://simrcompbio.shinyapps.io/sex_tnseq_viewer/). Briefly, this app includes all data for coding genes that passed our site depth and gene insert number cutoffs. The figures produced are similar to those in Fig 2 but are automatically generated for a user selected gene. This app also integrates meiotic transcription data from Mata et al. [49]. This tool will allow the *S. pombe* community, and more broadly, the community interested in sexual reproduction to easily navigate this dataset.

## Discussion

Here we present a framework to allow direct experimental assessment of the genetic underpinnings of sexual reproduction in an organism in a single internally controlled experiment, without use of pre-existing resources like an arrayed deletion collection. Our assay identified 532 genes as important for sexual reproduction, including 142 without a relevant previously annotated phenotype in *S. pombe* (S1 Table). 98 of these novel candidate genes are conserved in vertebrates, suggesting that our approach could provide a powerful tool to identify unknown but conserved sexual reproduction genes and guide candidate-based approaches in systems where high-throughput approaches like TN-seq are not practical.

Like previous assays, there are both advantages and drawbacks to the TN-seq approach. Our assay was able to test a similar number of genes to those that have employed the *S. pombe* deletion collection (3418 genes here vs 3285 genes [17]). The set was not completely overlapping, in part because our statistical approach had limited power for shorter genes with small numbers of unique inserts. One strength of this approach is that largely neutral intergenic sites serve as a built-in control for wild type sexual fitness. This is particularly important in *S. pombe* because we know from past work that wild type meiosis is already surprisingly error-prone [64]. In addition, while our statistical approach in this study utilized the existing *S. pombe* annotation, the transposon insertion itself is annotation independent. Thus, future work exploring the subset of noncoding inserts with strong fitness defects may illuminate the contribution of noncoding regions of the genome to sexual reproduction in a way that would be impossible with a deletion collection approach. In fact, while our study did not systematically explore noncoding inserts, inserts into the noncoding RNA *sme2*, which is involved in pairing of homologs during meiosis [71], were also underrepresented after sexual reproduction (Mann-Whitney U test, p = $1.945 * 10^{-12}$) (S15 Fig).

One additional strength of the TN-seq approach is that even in species with existing deletion collections like *S. pombe*, secondary screens can be performed that would otherwise be impractical. For example, to further categorize the role of hits from our TN-seq approach, we

could construct Hermes-insert libraries in a *rec12Δ* mutant background and repeat our assay. Genes that are required only to repair meiotic double-strand breaks should no longer have a phenotype in a *rec12Δ* mutant, implicating them in the repair of DSBs [72]. In contrast, some genes may be synthetically required for meiosis in the absence of *rec12*, suggesting a role in *rec12*-independent chromosome segregation [73]. This process could be repeated for mutants with defined phenotypes throughout meiosis to generate a systematic genetic map of meiosis in *S. pombe*. In contrast, with traditional methods this would require reconstructing the full deletion collection on multiple genetic backgrounds, some of which already have low mating efficiency and thus would be very difficult to construct via mating. Further, TN-seq could be employed to model human infertility alleles in their orthologs in fission yeast and reassay the network of genes required for fertility. This would allow powerful interrogation of allele-specific gene network interactions for human-relevant conditions. TN-seq opens up new experimental avenues even in traditional model organisms that have the potential to provide powerful new insights into previously studied pathways.

## Comparison with past screens

Our assay varied in approach from previous screens attempting to identify genes required for sex and meiosis using the *S. pombe* deletion collection. The first screen utilized iodine staining to specifically search for mutants that affected spore formation [40]. They identified 34 total genes spread over four categories, including genes required for zygote formation, iodine-reactivity, entry into meiosis, and for forespore membrane formation. Our screen assayed 29 of these genes and 17 were hits in our assay as well (58.6%). Ucisik-Akkaya et al. also identified a gold standard list of 13 known sporulation genes in *S. pombe* that had been experimentally tested. Their screen recovered 10 of the 11 they tested, while our assay found 11 of the 12 we tested. Interestingly, we also identified a mutant that produced inviable spores that nonetheless stained with iodine. This suggests that iodine staining screens can miss genes that make inviable spores. This observation is not surprising, given that this has previously been observed for mutants that produce a limited number of inviable spores, such as *meu10* [74].

In contrast, Blyth et al. cytologically examined meiotic chromosome segregation and sporulation via microscopy [17]. They scored two criteria: percentage of yeast cells that went on to produce spores (% sporulation) and percentage of tetrads that formed four spores, each with a single chromosome II lacI-GFP dot and defined mutants as <20% sporulation (n = 274) and/ or <90% four spore tetrads (n = 253). We recover a significantly larger proportion of Blyth et al.'s identified sporulation mutants (27.7%) than we do segregation mutants (15%) (Fisher Exact test, p = 0.0005). This discrepancy is likely at least in part because our assay looks at the endpoint of meiosis: the production and fitness of spores. Because *S. pombe* has only 3 chromosomes, a simple model of random chromosome segregation predicts that spore viability remains relatively high [73]. At Blyth et al.'s 90% cutoff for four spore tetrads, spore viability is still predicted to be greater than 90%. This level of segregation defect will result in only a relatively subtle effect on fitness and thus on observed frequencies via TN-seq, resulting in limited ability to detect mutants with very subtle phenotypes. For example, *hop1* contributes to meiosis in *S. pombe*, but a *hop1* deletion has only a mild effect on spore viability (89% of the wild-type fertility) and was not a significant hit in our assay [75]. Similarly, spore fitness can be affected by many processes other than failure to sporulate or chromosome segregation- *plb1* and *alg9* were both important, for growth post-sporulation for example.

The final screen also examined the *S. pombe* deletion collection but instead performed visual inspection of mating defects [41]. Dudin et al. identified 543 mutants with defects in mating or fusion behaviors on a 1 to 10 scale, with all scores described as mutant phenotypes.

Of those, we assayed 484 and recovered 83 (17.1%). Dudin et al. also scored low mating efficiency on a 1 to 10 scale. We recovered 53 of their 274 hits we tested (19.3%).

We also compared our data to a study examining genes that may be expressed during spore germination [43]. We assayed 117 of the genes highlighted in that study and found that 22 of those genes (18.8%) were functionally important for sexual reproduction in our assay.

We also note that our statistical approach used highly conservative multiple testing correction. *atg11* was a borderline non-hit in our screen after multiple testing correction but proved to have subtle sexual reproduction defects when we examined it carefully. This suggests that our assay may have limited power to detect subtle phenotypes and that there are likely additional components of the sexual reproduction machinery left to be identified in *S. pombe*.

## Sexual reproduction TN-seq can identify mutants with impaired gamete health

In contrast with previous approaches, our assay encompassed the entire process of sexual reproduction from mating all the way to spore germination. As a result, we were able to spot phenotypes not frequently observed in *S. pombe*. Typically, spore germination is measured by assessing the fraction of spores that grow into colonies. This reduces spore viability to a binary: spores either germinate or they do not. In contrast, our *plb1Δ* and *atg11Δ* mutants displayed a spore-specific growth defect where they germinated slowly but once germination began, they generally grew normally otherwise. *alg9Δ* mutants did exhibit a vegetative slow growth phenotype, but also had a severe delay in spore germination.

Past experiments have demonstrated a similar delay in spore germination in smaller *S. pombe* spores. While small spores are delayed in reaching polarization, once they do polarize, they go on to divide after a similar period of polarized growth as larger spores [76]. Thus, making smaller spores will likely lead to less competitive growth in a mixed environment with cells that make larger spores. While *plb1Δ* mutants made on average smaller spores than wild type, *alg9Δ* mutants did not and both strains had similar defects in spore outgrowth rate. In addition, *plb1Δ*, *alg9Δ*, and *atg11Δ* mutants exhibited a delay in initiation of isotropic growth but not a longer period of isotropic growth, suggesting that perhaps these mutants act in the process of breaking dormancy (reviewed in [77]) rather than simply making smaller spores.

There are likely additional genes whose mutants will exhibit similar defects. Mutants of *ntp1*, which is also involved in sexual reproduction via our assay, have been previously described to display delayed spore germination and delayed spore outgrowth [78] with otherwise normal vegetative growth rates [79]. This phenotype is also similar to that of *isp2Δ* mutants in the human fungal pathogen *Cryptococcus deneoformans*, which exhibit slowed growth after germination [80]. Similarly, cyclic AMP/PKA signaling has previously been reported to be involved in spore germination [81] and our TN-seq data confirms that both *pka1* and *cyr1* are involved in sexual reproduction.

More broadly, spore health can potentially model gamete health in mammals, as both spores and eggs/sperm are products of meiosis. Declining gamete health is one of the main drivers and most poorly understood aspects of age-associated human infertility [84]. While we have no evidence that *plb1Δ* or *alg9Δ* mutant spores decline in health over time, the underlying challenges in building a gamete that is stable for long periods of time may be similar between fungal spores and mammalian eggs. One primary difference is that mammalian oocytes have not yet completed meiosis and chromosome segregation, while fungal spores have. Thus, this assay may be more likely to identify components of chromosome segregation-independent mechanisms that are specifically required to recover from meiosis and return to active mitotic growth. Future experiments utilizing TN-seq to better understand these mechanisms in fission

## TN-seq can be used to search for density dependent growth mutants

In this study, we identified a gene (*sdg1*) that was previously annotated as essential but scored as both non-essential and involved in sexual reproduction in our assay. Through analysis of deletion mutants, we found *sdg1* to be non-essential, but required for growth at low density. This phenotype likely explains why this gene was originally misidentified as essential. Traditionally, tests of gene essentiality in fungi involve growth at low density. Tetrads from a heterozygote are dissected and a ratio of 2 live to 2 dead spores, where the live spores are wild type, is used to infer that the gene has an essential function. However, this test requires the ability to grow at low density from a single spore to produce a visible colony. Therefore, mutants like *sdg1*Δ that are unable to grow at low cell density may incorrectly appear essential in this test. Indeed, when plated at low density, *sdg1*Δ spores failed to grow the vast majority of the time (Fig 6A).

More generally, most tests of spore viability also employ either tetrad dissection or random spore analysis. Both tests require spores isolated on plates to germinate and form countable colonies. A mutant with a defect in low density growth would fail to successfully form a colony and would be mistakenly counted as an inviable spore. Thus, our results suggest that it is possible that a subset of other mutants annotated with spore viability defects may instead be mischaracterized low-density growth mutants.

Further, this result, in conjunction with earlier studies of auxotrophy, suggests that *S. pombe* has a density-dependent growth program [69,70]. Auxotrophs fail to grow at low density in the presence of ammonium or glutamate but grow at high density. Cells lacking *sdg1* fail to grow at low density but grow well at high density or in conditioned medium. While the defects caused by each appear independent, it remains unclear whether rescue of low-density growth is occurring through the same pathway in both scenarios. Future experiments can test whether there is one high density growth plan or multiple. We suggest that *sdg1*Δ mutants may represent activation of different or specialized genes to perform the same function at low and high densities (Fig 7D). There are 25 additional genes in the *S. pombe* genome with Acyl-CoA N-acyltransferase domains like *sdg1*, five of which are annotated as essential genes and could be candidates to perform the role of *sdg1* under a high-density growth program. Alternately, the unknown Sdg1-dependent function could be achieved by multiple redundant genes at high density or even not be required under a high-density growth plan. Mutants with similar quorum suppressible low-density growth defects exist in other species, such as *tup1*Δ mutants of *Cryptococcus deneoformans* [83], suggesting that low density growth plans may exist in many fungi.

The *sdg1*Δ phenotype also suggests another use of our TN-seq assay to directly identify mutants with low density growth defects. In our assay, our library of transposon-containing cells is plated at very low density so that they must form a microcolony in order to mate. Mutants that are capable of growth in dense culture, but are either incapable of forming a microcolony, or form smaller microcolonies, will fail to pass through our sexual reproduction assay. This is technically a false positive for our sexual reproduction assay, but a similar approach could be used specifically to identify genes involved in low density growth. This low-tech sparse plating approach may achieve the same goals as the more complex Droplet Tn-seq microfluidics and single cell amplification method recently described [82].

### Exploring other members of the *Schizosaccharomycetes* clade and related species

We anticipate that our TN-seq approach will allow highly productive future exploration of the evolution of sexual reproduction in the *Schizosaccharomycetes* lineage. Exploration of this complex could provide a model to study the evolution of the genetics of sexual reproduction over time, which is particularly interesting given the presence in *S. pombe* of a family of meiotic drivers called *wtf* genes [62,85–86] that appear to generate a selective pressure for less efficient meiosis [64].

In addition, a pan-genus understanding of sexual reproduction and meiosis in the *Schizosaccharomycetes* could provide valuable insight into the biology of *Pneumocystis jirovecii*, a fungal pathogen of humans that is a sister to the *Schizosaccharomycetes* lineage. *P. jirovecii* is responsible for 400,000 annual life-threatening infections, typically in conjunction with HIV/AIDS [87]. Because *P. jirovecii* cannot be grown as a free-living organism, studies in related organisms will be critical to understand its biology. This is particularly relevant as recent studies have suggested that *P. jirovecii* may be undergoing sexual reproduction inside human hosts in order to produce sexual spores that can be transmitted to a new host to establish infection [88–90]. Therefore, understanding spore formation and germination, as well as developing inhibitors for it, could be critically useful in blunting *P. jirovecii* infection. More broadly, spores often serve as infectious particles for many fungal pathogens [91] and thus understanding the genetic network required to produce functional spores could be helpful in preventing infection.

## Material and methods

### Media

We made YEA+SUP plates with 5 g/L Bacto Yeast Extract, 30 g/L Dextrose, 20 g/L Bacto Agar, 250 mg/L Adenine, 250 mg/L Histidine, 250 mg/L Leucine, 250 mg/L Lysine, 250 mg/L Uracil. We made YEL+SUP in the same way but omitting the Bacto Agar. Media for G418 selection was routinely supplemented with 100 mg/L G418 and media for hygromycin selection was routinely supplemented with 200 mg/L hygromycin.

We made *S. pombe* conditioned medium by starting a 5 mL YEL+SUP culture from a single colony of SZY643 ($h^{90}$, *leu1⁻*, *ura4⁻*). We grew that culture overnight at 32˚C and diluted it back into 50 mL of fresh YEL+SUP. We then grew that culture overnight as well and then diluted it back into 500 mL of YEL+SUP and grew it overnight again. We then collected that entire culture via centrifugation at 5000g for 10 minutes at 4˚C. We filtered the supernatant through a 0.2 μm vacuum filter and stored it at 4˚C until use. We then made conditioned medium using the standard YEA+SUP recipe, except that we replaced 50% of the water with the filtered culture supernatant described above prior to autoclaving. We stored plates at 4˚C in sealed plastic sleeves until use.

### Strain construction

We transformed all *S. pombe* strains using a standard lithium acetate transformation protocol [92]. We constructed all mutants with at least two independent transformations using independent starting cultures and we then PCR validated to confirm correct insertion of the deletion cassette and loss of the original coding sequence. Strains are listed in S3 Table. We cryopreserved all strains at -80˚C in 20% glycerol and revived them on yeast extract agar (YEA+SUP) at 32˚C prior to conducting experiments. For deletion of *ifs1Δ*, *atg11Δ*, *plb1Δ*, and *alg9Δ*, we amplified the *kanMX4* allele from the *S. pombe* deletion collection strain with

approximately 500 base pairs of homology up and downstream. Primers for each can be found in S4 Table. For deletion of *sdg1Δ* and *ggt1Δ*, we constructed deletion cassettes via overlap PCR with *kanMX4* from plasmid pFA6 [93]. Primers for these overlap PCR reactions are also listed in S4 Table. We were able to successfully delete *sdg1Δ* without modification to our lithium acetate transformation protocol. We speculate that these transformants were able to grow on the initial transformation plates despite the low-density growth defect because of quorum signals that accumulated in the non-selective recovery medium and were then spread onto the selective plates.

We constructed prototrophic *sdg1Δ* mutants by crossing our auxotrophic *sdg1Δ* mutants to strain SZY513 and collecting the spores. We then plated and grew them on YEA+SUP+G418 (100 mg/L) to select for *sdg1Δ::kanMX4* and then replica plated to minimal medium to select for prototrophs (for SZY4831). Alternatively, we initially grew them on minimal medium and then replica plated to YEA+SUP+G418 (for SZY4834 and SZY4837).

To generate our *ifs1Δ* strain in an Isp3-GFP background (SZY5082), we transformed the *ifs1::kanMX4* deletion cassette into SZY4975 ($h^{90}$ *isp3*-GFP *leu1*::mCherry-psy1: FY39320 from the Yeast Genetic Resource Center) [94].

## TN-seq library preparation

We prepared transposon insertion libraries using a modified method from the one presented in Guo et al [30]. We transformed plasmid pHL2577 (*URA3*, *KANMX6*) carrying the Hermes transposon and pHL2574 (*LEU2*) carrying the NMT-transposase into SZY643 ($h^{90}$, *leu1*⁻, *ura4*⁻) and selected transformants on EMM+ 250 mg/L adenine + 250 mg/L histidine+ 250 mg/L lysine+ 10 mM Thiamine medium. Rather than picking individual colonies, we scraped the transformation plate and collected all transformants to inoculate a 50 mL EMM+adenine +histidine+lysine+Thiamine culture to grow overnight until saturated. We then washed 3 times with EMM+adenine+histidine+lysine (lacking thiamine) to remove any residual thiamine and resuspended in 25 mL of EMM+adenine+histidine+lysine. We used 500 μL to inoculate 50mL of EMM+adenine+histidine+lysine to induce transposition. We then serial cultured over 4 days by adding 1 mL (day 2) or 1.5 mL (days 3 and 4) to 50 mL of fresh EMM +adenine+histidine+lysine. We selected for cells with transposition events by first spinning down 10 ml of this final culture (3000 rpm, 5 min), removing 9 mL of the supernatant and resuspending the cells in the remaining 1 mL of supernatant. We used this to inoculate 100 mL of YEL+SUP+5-FOA (1 g/L) and grew overnight to counter-select against the initial Hermes plasmid. We then spun 50 mL down as previously described and resuspended the cells in 5 mL of the supernatant. We inoculated this into 500 mL of YEL+SUP+5-FOA (1g/ L) +G418 (500 mg/L) and grew it for 2 days until saturation. The resulting *ura*⁻ and G418ᴿ cells should have undergone transposition but lost the pHL2577 plasmid. We collected cells and split them to freeze in 20% glycerol as well as to prepare DNA.

We prepared DNA using a Qiagen Genomic Tip column with the standard yeast protocol except we extended the lyticase and proteinase K treatments to 24 hours on a shaking incubator. We then digested the purified DNA using either MseI (10,000 U/mL) or a combination of HpaII (10,000 U/mL), TaqI-V2 (20,000 U/mL), and HpyCH4IV (10,000 U/mL). Our digests used 10 μg of DNA in 500 μL of volume overnight at 37˚C in CutSmart buffer with 15 μL of each enzyme, except that TaqI-V2 was not added until after the overnight digest, after which we raised the temperature to 65˚C for 4 additional hours. We cleaned and size-selected the digested DNA using SPRIselect beads by washing with 0.5 volumes of SPRIselect beads, pelleting on a magnet and retaining the supernatant, and then adding 0.2 volumes of beads and

precipitating on beads again. Finally, we washed and eluted the DNA from the beads with 500 μL of water.

We then end-ligated on linkers that contained unique, random barcodes using a reaction containing 490 μL of the cleaned, size-selected DNA, 88 μL of 10x ligation buffer, 143.5 μL of water, 153.5 μL annealed barcoded linker, and 5 μL T4 DNA ligase (400,000 U/mL). We then divided the reaction evenly between 32 PCR tubes and incubated for 16 hours at 16˚C. We prepared the barcoded adapter as in Guo et al. [30] using oligo oSZ2760 that contains random nucleotides for the unique barcode, and either oSZ2485 for the MseI digested DNA or oSZ2499 for the combination HpaII, TaqI-V2, and HpyCH4IV digested DNA (S4 Table). We annealed the oligos together by mixing at a concentration of 10 μM each in 1x HF PCR buffer and denatured at 95˚C for 1 minutes, followed by decreasing the temperature 10˚C for 7 minutes, and then further decreasing the temperature by 10˚C every 7 minutes until it reached 20˚C. We stored the annealed adapter at -20˚C. The unique barcodes allow us to directly count the number of inserts present in our library after sequencing [33].

At this stage, we split the DNA from each digest into two pools (four pools total). We amplified these pools via PCR with Phusion polymerase and 15 different Hermes-specific oligos (oSZ2104-2118, S4 Table) with Illumina adapter tails. We used 3 reactions per oligo per pool, resulting in a total of 90 PCR reactions for each enzyme mix. In addition, we ran 3 linker-only controls. Each PCR reaction contained 24.5 μL water, 10 μL 5x HF buffer, 1 μL dNTPs (10 mM), 8 μL linker ligated DNA, 1 μL linker oligo oSZ2483 (10 μM), 5 μL Hermes-specific oligo (2 μM) or water, and 0.5 μL Phusion polymerase (2,000 U/mL). We amplified inserts using the same thermocycler settings as Guo et al. [30]: 94˚C for 1 min, 6x (94˚C for 15 sec, 65˚C for 30 sec, 72˚C for 30 sec), 24x (94˚C for 15 sec, 60˚C for 30 sec, 72˚C for 30 sec), 72˚C for 10 min, 4˚C hold. For each primer pair and pool, we then combined the three reactions to form sub-pools, a subset of which we ran on a gel for validation. We then added the remaining Illumina adapters and barcodes with another round of PCR. This PCR reaction contained 28.5 μL water, 10 μL 5x HF buffer, 1 μL dNTPs (10 mM), 2.5 μL oSZ2128 (10 μM), 2.5μL barcode oligo (oSZ2129-2136, 2763–2766; 10μM), 5 μL of insert PCR pool, 0.5 μL Phusion polymerase (2,000 U/mL) and we incubated it at 94˚C for 2 min, 5x (94˚C for 30 sec, 54˚C for 30 sec, 72˚C for 40 sec), 4˚C hold. The individual pools are given a unique barcode (2 barcodes per enzyme digest mix and 4 total barcodes per DNA library). We then cleaned the DNA using 0.75 volumes of SPRI beads, precipitated on a magnet, washed, and eluted. We confirmed the DNA library by PCR using oligos to the ends of the Illumina adapters. This reaction consisted of 15.75 μL water, 5 μL 5x HF buffer, 0.5 μL dNTPs (10 mM), 1.25 μL oSZ2176 (10 μM), 1.25 μL oSZ2177 (10 μM), 1 μL library DNA, 0.25 μL Phusion polymerase (2,000 U/mL). We ran this PCR at 98˚C for 30 sec, 35x (98˚C for 10 sec, 62˚C for 30 sec, 72˚C for 40 sec), 72˚C for 10 min, 4˚C hold. Finally, we quantified this DNA library using a Qubit and determined fragment sizes using a Bioanalyzer. Sequencing was performed on a NextSeq instrument at the Stowers Molecular Biology Core. Raw sequencing reads are available at PRJNA758956 on the Sequence Read Archive.

## TN-seq sexual reproduction assay

We revived a frozen aliquot (1.8 mL) of our *S. pombe* Hermes TN-seq library by thawing it on ice and growing it in 50 mL of saturated culture in YEL+SUP overnight at 32˚C. We used part of this culture to prepare a second round of pre-sex sequencing libraries (see above), while we diluted the remainder 1000-fold for plating. We plated 250 μL of this diluted culture (an estimated 18,500 cells) to each of 144 large (150 mm) MEA plates. We then incubated these plates at 25˚C for 9 days. After incubation, we collected cells from the MEA plates by physically

scraping cells from all 144 plates. We specifically isolated spores using a standard glusulase prep [59] and then allowed them to germinate in YEL+SUP liquid culture for either a "short" (single 1:100 dilution of spores into rich medium and grown for 24 hours) or "long" (1:1000 dilution of spores into rich medium grown for 24 hours and diluted 1:100 into rich medium and grown for another 24 hours) outgrowth period. We then collected cells and isolated DNA using the Qiagen Genomic Tip kits. This DNA was used to prepare sequencing libraries as described above.

## Data analysis

We initially processed sequencing reads using a custom R script to identify reads with the correct linker sequence. We kept only reads with a perfect match to the linker sequence. We saved and linked the unique barcode sequence added with the linker to the read while the entire linker sequence was trimmed off. We then imported reads with the proper linker sequence into Geneious Prime (v. 2020.2). We removed reads if they did not contain sequence matching perfectly to the entire end of the Hermes transposon from oligo to insertion site, 61–85 bp depending on the oligo. We did this using the "Separate Reads by Barcode" function in Geneious which also trims the Hermes sequence from the read at the same time. We mapped the remaining reads to the *S. pombe* genome (version ASM294v2) using the Geneious aligner within Geneious. Non-unique mappings were not allowed. We then output this alignment and ran it through a custom R script to generate a list of read start positions (i.e. Hermes insert sites) with a corresponding depth of unique barcodes at that position for each sequencing library. We used custom Perl scripts to assign genes to each insert site based on the *S. pombe* annotation (version ASM294v2), to combine depth files from the 4 individual sequencing runs for each treatment, and to track positions where inserts were present in the initial library but missing from subsequent libraries. We then used R to perform statistical tests. To limit low-frequency sampling issues, we dropped any site with eight or less unique barcoded ligation products in our pre-sex dataset. We calculated a $\log_{10}$ fold change for post-sexual reproduction over pre-sexual reproduction frequencies. To perform this analysis, we set all the detected number of reads for all 0 sites in the post-sex libraries to 1. This artificially reduces the magnitude of phenotypes observed but allows log adjustment of our data. We then filtered out all genes with less than 5 insert sites. We used a Mann-Whitney U test to compare the distribution of fold-changes for the remaining genes to that of the inserts annotated as intergenic. Finally, we performed multiple test correction using a Bonferroni correction.

## Viable spore yield

We started overnight liquid cultures from a single colony in 5 mL of YEL. The next day, we spotted 50 μL saturated culture in a single patch (for dense patches) or spread 111 μL of a $10^{-3}$ dilution onto an entire SPAS or MEA plate (for sparse patches). The sparse dilution is scaled to reproduce the same plating density used in the initial TN-seq assay but on a smaller petri plate. At the same time, we made serial dilutions of these cultures and plated to YEA+SUP plates to determine the number of colonies per mL and thus the number of colony forming units we had spotted on the mating plates. We incubated the mating patches at 25˚C. After either 3 or 9 days, we scraped the entire patches off the plates and used a standard glusulase prep to isolate spores [59]. We then performed a serial dilution and plated spores to YEA+SUP plates in order to count the number of spores present in that patch. We calculated viable spore yield by dividing the number of viable spores present in the patch by the number of yeast cells originally plated. Additionally, we have normalized each experiment to the mean VSY of a control (SZY643) experiment done at the same time.

## Iodine staining

We spotted 100 μL of culture from saturated overnight liquid cultures onto MEA plates and incubated them for 3 days at 25˚C. We then inverted these plates over iodine crystals in a chemical fume hood until positive control cell patches had stained. We scored and imaged these plates immediately.

## Aneuploidy assay

We generated test strains to measure rate of aneuploidy for chromosome 3 by crossing our *plb1Δ*, *atg11Δ*, and *alg9Δ* mutants to a strain (SZY2465, *ade6Δ::hphMX6*, *h⁻*) carrying *ade6* deleted with *hphMX6*. We grew overnight cultures of each strain in liquid YEL+SUP medium at 32˚C. We then mixed equal volumes of each parent strain and spotted them to MEA. We incubated these plates for 3 days at 25˚C. We then collected spores, glusulase treated, and plated spores to YEA+SUP+Hyg plates. We picked and master plated spore colonies to YEA+SUP plates. We then replica plated to identify progeny that were *ade6Δ::* hphMX6, *ura4⁻*,*h⁻*. We also generated a second test strain by crossing our *plb1Δ*, *atg11Δ*, and *alg9Δ* mutants to an *h⁺* strain SZY128 (*ade6⁻*, *leu1⁻*, *his5⁻*, *h⁺*). We carried this assay out as described above except that spores were plated to YEA+SUP rather than YEA+SUP+Hyg. From this cross, we selected progeny that were *leu1⁻*, *his5⁻*, $G418^R$, *h⁺*. From each cross, we also selected a strain with the same genotype but with a wild type allele of *plb1*, *atg11Δ*, and *alg9*.

To assay aneuploidy, we crossed our *h⁺* and *h⁻* mutant strains to each other and our *h⁺* and *h⁻* wild type strains to each other on MEA for 3 days at 25˚C. We collected spores, glusulase treated, diluted them, and plated on YEA+SUP plates. We picked entire spore colonies to make a YEA+SUP masterplate after 5 days. This plate was replica-plated to each test medium and chromosome 3 disomes were scored as colonies that were both ade+ and hygromycin resistant. We used the remaining markers to verify that strains were mating as expected (approximately 50:50 inheritance of unlinked markers). We did not score crosses with obvious strong divergence from 50:50 inheritance of the unlinked markers.

## Competition assay

To conduct spore competition assays, we mixed approximately 100,000 viable parental spores (*h⁹⁰*, *leu1⁻*, *ura4⁻*) with either 100,000 spores of our tester mutant strain or 100,000 spores of a presumably neutral *wtf12Δ* pseudogene deletion that was also G418-resistant. Spore numbers were quantified by plating cells on YEA+SUP and determining the colony forming units. We then diluted the spore mix into 5 mL of YEL+SUP, took a sample and plated to determine the initial frequency. We replica plated these plates to YEA+SUP+G418 to count the proportion of G418-resistant and susceptible colonies. We also grew the 5 mL mixed culture overnight with shaking. The next day, we diluted and plated these spores to YEA+SUP to acquire approximately 200 colonies per plate. Once they had grown, we replica plated these plates as well to determine the final frequency of each strain in the mix.

For vegetative competition assays, we started 5 mL overnight cultures using single colonies of the parental strain, our test mutant, and the *wtf12Δ* deletion strain. The next day, we determined $OD_{600}$ for these cultures using a spectrophotometer and diluted them back to an $OD_{600}$ of 0.001. We then mixed each with an equal amount of adjusted parental strain in 5 mLs of fresh YEL+SUP. We sampled and plated this initial mix, as above, and then grew the competition cultures overnight with shaking. Finally, we plated and replica plated to determine the proportion of each strain found in the initial and final mix.

## *S. pombe* functional annotation

We acquired *S. pombe* functional annotations (FYPO) and GO annotations from PomBase on August 7, 2022 [48]. To compare to our TN-seq data, we selected only annotations representing deletion phenotypes. We then compared insert fold changes within the set of genes with a given annotation to the background set of all genes lacking that annotation. We tested the difference between these means using a Mann-Whitney U test.

For annotations of hits, we selected manually selected relevant FYPO terms relevant to sexual reproduction (see S1 File). We selected GO terms by picking parent terms (Mating-type Determination, Meiotic cell cycle, Sexual Reproduction) and selecting all descendant GO terms using GO.db [95]. We then searched our list of terms against the PomBase annotations using custom perl scripts. We took the same approach for assigning metabolic and biosynthetic annotations, but with GO terms and their descendants for generation of precursor metabolites and energy, lipid metabolic process, carbohydrate metabolic process, ribosome biogenesis, translation, tRNA metabolic process, and protein modification process, as well as deletions annotated with auxotrophy phenotypes (FYPO:0000128 and descendant terms).

## Spot dilution assays

To perform spot dilution assays, we grew cultures overnight in YEL until they reached saturation. We then measured the $OD_{600}$ using a spectrophotometer. We pelleted enough culture to produce an $OD_{600}$ of 20 in a final concentration of 1 mL by spinning at 3000 rpm for 5 min. We then removed the supernatant by pipetting and resuspended in 1 mL of water to remove any effects of conditioning in the residual medium. We then performed 10-fold serial dilutions to a final concentration of $OD_{600} = 2*10^{-4}$. We spotted 5 μL spots of each intermediate dilution onto the appropriate medium and incubated plates at 32˚C, or 25˚C for MEA medium, until we took pictures as detailed in individual figures.

## Microscopy

**CellASIC imaging.** We imaged spore germination using a cellASIC ONIX microfluidic plate (YO4D-D2-5PK) and CellASIC ONIX2 Software. We washed and flushed the microfluidic plates with PBS prior to loading with YEL. We then added 100 uL of spores to cell collection wells and ran the cell loading sequence to trap spores in the imaging chamber. After loading, we incubated the plates at 32˚C with flowing YEL medium and imaged every 10 minutes for 48 hours on a Nikon Widefield Ti2 microscope with a 60x objective. For each video, we obtained two XY locations per channel.

**Manual tracking of spores.** We first used Fiji to convert.nd2 files from the time lapse into tifs. We then manually drew regions of interest (ROIs) around the spore wall for all spores in a field of view that went on to grow and divide during the course of the video. We tracked the start of swelling by keeping the ROI centered over the spore and moving forward frame by frame until the spore grows larger than the ROI. This frame was recorded as the "swelling" point. We then manually tracked the first visible point where one side of the spore begins to polarize (ie. the spore becomes asymmetric), as well as the times of the first, second, and third divisions.

**Agar punch live imaging of spore germination.** To perform live imaging, we spread isolated spores onto a YEA+SUP plate using beads. Immediately after the plates dried, we removed the beads from the plate and removed a circular punch of agar from the plate using a 1271E Arch Punch as previously described [96]. We inverted the agar and placed it top side down into a 35 mm glass bottom dish (No. 1.5 MatTek Corporation) containing a damp kimwipe and sealed the lid with vacuum grease. We then mounted the dish in a stage top incubator

(Oko Lab) to maintain a temperature of 32˚C. We imaged continuously on a Nikon Ti2-E widefield microscope, taking brightfield exposures using a Plan Apochromat Lambda 60x objective (1.4 NA) every 10 minutes for either 24 or 48 hours total. We processed images with Fiji (https://imagej.net/software/fiji/) to remove drift using stackregj (when necessary) and cropped to smaller image sizes for convenience. We manually scored steps throughout spore germination and division using Fiji and by manually drawing ROIs.

**Still imaging of spores for bright field and fluorescence microscopy.** To take still images of cells sporulated on MEA we scraped a small portion of cells from the plate, resuspended in lectin (1 mg/mL) and imaged on a slide. To image cells grown on SPA, we used the punch method described above. Cells were imaged using an Axio Observer.Z1 (Zeiss) wide-field microscope with a 40x C-Apochromat (1.2 NA) water-immersion objective. To excite GFP, we used a 440–470 nm bandpass filter, reflected the beam off an FT 495 nm dichroic filter into the objective, and collected emission using a 525–550 nm bandpass filter. We collected emission onto a Hamamatsu ORCA Flash 4.0 using μManager software.

## Deep learning to identify and track spores

We segmented images using the Mask R-CNN [97]convolutional neural network as implemented in the pytorch [98] model zoo. Mask R-CNN is a neural network pre-trained to identify and segment everyday objects from a large set of images. Since the Mask R-CNN network was not trained specifically on cells, we fine-tuned it to identify *S. pombe* cells and spores. We did this by manually outlining or filling spores and cells in 12 transmitted light images and then retraining the network. During training, we sliced images into random 400x400 patches and augmented this with rotation and mirroring every iteration. We ran training for 200 epochs. We performed inference by slicing an input image into 400x400 patches, running the network on each patch, and reassembling the predicted patches back to the same size as the original image. The training and inference packages were written in python using pytorch as the deep learning library. We performed computation on a workstation with Ubuntu Linux 18.04 and an NVIDIA Quadro RTX 8000 for a GPU.

We loaded the output (predictions) of the deep learning network into Fiji (https://imagej. net/software/fiji/) for analysis. First, we thresholded the predictions manually to create a binary mask. We then registered this mask for movement when necessary, using "Stackregj." Next, we turned this mask into a list of ROIs using "Analyze Particles." We measured these ROIs in order and acquired all Fiji measurements. These measurements included, but were not limited to, the area, the aspect ratio, and the minimum and maximum axes of a fit ellipse, among others. We saved these measurements as a.csv file and loaded them into a jupyter notebook written in-house to track spores from one frame to another.

## Supporting information

**S1 Fig. Insert distribution is similar to published data.** Histograms of insert density per gene both from published data [30] (A) and our data (B). Both plots are normalized with the number of unique insertion sites per gene normalized to the length in kilobases and per one million inserts. Genes annotated essential in Guo et al. [30] are in pink (top) and genes annotated non-essential are in green (bottom).
(TIF)

**S2 Fig. Insert number but not distribution is altered by filtering.** Genome-wide histograms of unique insert density before (A) and after (B) filtering to remove inserts with fewer than 8 unique ligation products in our sequencing. A 1 megabase region of chromosome II is shown

as an inset. While filtering reduces the total number of inserts in a window, it does not substantially change the relative insert density.
(TIF)

**S3 Fig. Candidate repressors largely appear to have growth rate advantages.** TN-seq insert frequencies for 15 genes identified as candidate repressors of sexual reproduction. The y-axis displays the $\log_{10}$ adjusted ratio of mean insert frequency across a gene to the mean insert frequency for that gene at the first sequencing step. The first and third segments are vegetative growth, while the second is a sexual reproduction step. Genes whose insert frequencies did not increase at every step are shown with dashed lines.
(TIF)

**S4 Fig. Candidate mutant genes are broadly conserved in vertebrates and include a mix of known sex and unknown genes.** Venn diagrams showing the breakdown of candidate genes by conservation status (A) and by function (B). Both gene sets are broken down in S1 Table and both are derived from annotations on Pombase [66].
(TIF)

**S5 Fig. Candidate mutants produce sexual spores.** Imaging on an AXIO Observer.Z1 (Zeiss) wide-field microscope with a 40x C-Apochromat (1.2 NA) water-immersion objective of wild type and mutant *S. pombe* sexual spores/asci produced after 2 days at 25˚C on MEA. Scale bars indicate 10 microns.
(TIF)

**S6 Fig. Spores produced by *ifs1*Δ diploids are irregular and snowman-shaped in SPA medium.** Isp3-GFP was visualized in wild type and *ifs1*Δ mutants on an AXIO Observer.Z1 (Zeiss) wide-field microscope with a 40x C-Apochromat (1.2 NA) water-immersion objective after incubation for 2 days at 25˚C on SPA medium. Scale bars indicate 10 microns. Arrow indicates a "snowman" spore.
(TIF)

**S7 Fig. Surviving *ifs1*Δ spores retain their mutant phenotype.** Viable spore yield assay showing on the y-axis the number of spores produced per yeast cell plated, normalized to the mean value for wild type. Cells were incubated on MEA plates in a dense growth spot for 3 days at 25˚C prior to spore isolation. Results from two independent *ifs1*Δ mutants are displayed, as well as from two spores that germinated from independent biological replicates of the *ifs1*Δ #2 mutant. All mutants were assayed in a set of at least 5 biological replicates alongside at least 5 wild type replicates. Points display results from a single replicate, normalized to the mean from the corresponding wild type controls. The boxplots summarize the underlying points and show first quartile, median, third quartile while the whiskers show the range of the data to a maximum of 1.5 times the interquartile range below and above the first and third quartile, respectively. Points outside the whiskers can be considered outliers.
(TIF)

**S8 Fig. Viable spore yield varies by condition.** A-E) Viable spore yield assay showing on the y-axis the number of spores produced per yeast cell plated, normalized to the mean value for wild type. Points display normalized results from a single replicate. The boxplot summarizes the underlying points and show first quartile, median, third quartile while the whiskers show the range of the data to a maximum of 1.5 times the interquartile range below and above the first and third quartile, respectively. Points outside the whiskers can be considered outliers. A) Cells incubated on SPAS plates for three days at 25˚C in dense cell patches (Mann-Whitney U test, *alg9*Δ, p = 0.065; *plb1*Δ, p = 0.94; *ifs1*Δ, p = 0.0044). B) Cells incubated on MEA plates for

three days at 25°C at low density as in the original TN-seq assay (Mann-Whitney U test, *alg9Δ*, p = 0.0022; *plb1Δ*, p = 0.0022). C) Cells incubated on MEA plates for nine days at 25°C in dense cell patches. (Mann-Whitney U test, *alg9Δ*, p = 0.0043; *plb1Δ*, p = 0.18) D) Cells incubated on MEA plates for nine days at 25°C at low density. These conditions match the original TN-seq assay, except that spores were not germinated in liquid. (Mann-Whitney U test, *alg9Δ*, p = 0.0022; *plb1Δ*, p = 0.24), E) Summary data encompassing all conditions tested for these three mutants. This includes data from Fig 3D as well as S8A–S8D Fig (Mann-Whitney U test, *alg9Δ*, p = 0.0013; *plb1Δ*, p = 0.73; *ifs1Δ*, p = $1.1^*10^{-14}$; *atg11Δ*, p = 0.14).
(TIF)

**S9 Fig. *plb1Δ* mutant spores germinate more slowly in an agar punch imaging approach.**
A) *S. pombe* spores undergo several landmarks in the process of germinating. Spores initially begin growing isotropically (ie. swelling). This phase ends when cells begin polarized growth and elongate on one side. This cell will eventually divide by fission for the first time and each of those daughters will go on to divide a second time after some delay. We scored each of these landmarks manually from videos using Fiji. Unlike in our microfluidics approach, we were unable to score the delay to swelling or the third cell division with this approach. B) Spores were plated on YEA+SUP plates and a punch was immediately taken and imaged at 32°C for 24 to 48 hours. Videos of spore germination were scored and time between each step in spore germination was tracked for individual spores. The boxplot shows first quartile, median, and third quartile while the whiskers show the range of the data to a maximum of 1.5 times the interquartile range below and above the first and third quartile, respectively. Points outside the whiskers can be considered outliers. C) Histogram of spore sizes from the first 10 frames (100 minutes) of videos of spore germination. Wild type and *plb1Δ* are each derived from at least two videos each from two separate days. Spores were identified using deep learning (see methods). D) Plot of average cell area over the course of videos of spore germination. Spores were identified via deep learning. Data are derived from at least two videos each from two separate days. Average spore sizes stabilize once spores begin to divide and grow vegetatively as yeast cells. Data are truncated at 1500 minutes as dividing cells begin to affect the average in *plb1Δ* mutants. The average size of *plb1Δ* mutant cells does not reach wild type levels by the time cells have divided enough to make continued tracking impossible with this approach.
(TIF)

**S10 Fig. *plb1Δ* and *alg9Δ* mutants exhibit a delay in growth and in establishing polarity.** A) Schematic of plots of aspect ratio versus area. The y-axis displays aspect ratio, where a minimum value indicates round cells and larger values indicate more oblong cells. The x-axis displays normalized cell size. As shown in the cartoon, a higher aspect ratio indicates cells that are more elongated, while a higher area indicates cells that are larger. B) Two-dimensional histograms showing the entire population of spores for wild type or mutant in a given time window. Contour lines are added to help visualize concentration of cells.
(TIF)

**S11 Fig. Conditioned medium from different species provides varying levels of rescue for *sdg1Δ* mutants.** A-G) Spot dilution assays with 5 μL spots plated. The initial leftmost spot is of $OD_{600}$ = 20 culture and each successive spot is a 10-fold dilution, so that the final spot should be $10^5$ less concentrated than the first. All four experiments were conducted on the same day with the same dilution series of parent strain (*ura4*-D18, *leu1*-32) and three independent *sdg1Δ* mutants on the same genetic background (*sdg1Δ::kanMX4*, *ura4*-D18, *leu1*-32). All assays were also incubated for 4 days at 32°C. A) Spotted to standard yeast extract agar (YEA+SUP). B) Spotted to YEA+SUP where half the water had instead been replaced with yeast

extract liquid with supplements (YEL+SUP) medium as a control for conditioned medium. C) Spotted to conditioned YEA+SUP medium where half the water had instead been replaced with YEL+SUP pregrown with the parent strain *S. pombe* (*ura4*-D18, *leu1*-32) (see methods). D) Spotted to conditioned YEA+SUP medium where half the water had been replaced with YEL+SUP pregrown with an *sdg1Δ* mutant (*sdg1Δ::kanMX4, ura4*-D18, *leu1*-32). E) Spotted to conditioned YEA+SUP medium where half the water had been replaced with YEL+SUP pregrown with *S. octosporus*. F) Spotted to conditioned YEA+SUP medium where half the water had been replaced with YEL+SUP pregrown with *S. japonicus*. G) Spotted to conditioned YEA+SUP medium where half the water had been replaced with YEL+SUP pregrown with *S. cerevisiae*.
(TIF)

**S12 Fig. *sdg1Δ* does not cause auxotrophy.** A-B) Spot dilution assays with 5 μL spots plated. The initial leftmost spot is of $OD_{600}$ = 20 culture and each successive spot is a 10-fold dilution, so that the final spot should be $10^5$ less concentrated than the first. Both experiments were conducted on the same day with the same dilution series of parent strain (*ura4*-D18, *leu1*-32), three independent *sdg1Δ* mutants on the same genetic background (*sdg1Δ::kanMX4, ura4*-D18, *leu1*-32), a wild type prototroph ($h^{90}$), and three independent prototrophic *sdg1Δ* mutants. A) Spotted to standard yeast extract agar (YEA+SUP) and incubated for 4 days at 32˚C. Note that the top half of this panel is the same experiment presented in Fig 5C. B) Spotted to minimal medium and incubated for 4 days at 32˚C.
(TIF)

**S13 Fig. Ammonium chloride enhances low density growth defect of auxotrophic but not prototrophic *sdg1Δ* mutants.** A-B) Spot dilution assays with 5 μL spots plated. The initial leftmost spot is of $OD_{600}$ = 20 culture and each successive spot is a 10-fold dilution, so that the final spot should be $10^5$ less concentrated than the first. Both experiments were conducted on the same day with the same dilution series of parent strain (*ura4*-D18, *leu1*-32), three independent *sdg1Δ* mutants on the same genetic background (*sdg1Δ::kanMX4, ura4*-D18, *leu1*-32), a wild type prototroph, and three independent prototrophic *sdg1Δ* mutants. A) Spotted to standard yeast extract agar (YEA+SUP) and incubated for 4 days at 32˚C. B) Spotted to yeast extract agar with 5 g/L supplemental ammonium chloride and incubated for 4 days at 32˚C.
(TIF)

**S14 Fig. Sexual reproduction TN-seq data can be visualized using a publicly available Shiny app.** Screenshot of a publicly available interactive Shiny app that visualizes data from the TN-seq assay. There are five plots, displaying the distribution of insert fold changes within a gene (as in Fig 2B), the mean insert frequency within a gene over time (as in S3 Fig), the site-by-site frequency across a gene (as in Fig 2A), a volcano plot summarizing the entire experiment (as in Fig 2C), and a plot of transcription throughout meiosis for that gene from the Mata et al. dataset [49] (as in Fig 2D). This app will accept *S. pombe* gene names, either as systematic names or common names.
(TIF)

**S15 Fig. Inserts into the noncoding RNA *sme2* are significantly underrepresented after sexual reproduction.** Boxplot displaying distribution of $\log_{10}$-adjusted fold changes in insert density after sex (ie. frequency after sex/frequency before sex). Boxplots show first quartile, median, third quartile. The whiskers show the range to a maximum of 1.5 times the interquartile range above and below the first and third quartile, respectively. Outlier data points (outside the whiskers) are not displayed. This results in 5,716 of 235,578 intergenic sites, and 0 of 78 sites from *sme2* not being displayed although those data were considered in the statistical

analyses. Inserts in intergenic regions are indicated in grey and inserts into the known meiotic noncoding RNA *sme2* are shown in orange.
(TIF)

**S1 File. Selected sexual reproduction relevant phenotype annotations.**
(TXT)

**S1 Movie. *plb1Δ* and *alg9Δ* mutants exhibit a delay in growth and in establishing polarity.**
Video showing sequential two-dimensional histograms showing the entire population of spores for wild type or mutant in a given time window. Contour lines are added to help visualize concentration of cells.
(MOV)

**S1 Table. Candidate Genes Identified by meiotic TN-seq.**
(XLSX)

**S2 Table. Aneuploidy Assay.** S2 Table describes the results of three pairs of experiments conducted on different days. As a result, each mutant has a set of paired wild type data. Pooling the wild type data from different days does not result in a change in the result (Fisher's Exact Test; *plb1Δ*, p = 0.7926; *alg9Δ*, p = 0.165; *atg11Δ*, p = 0.7421).
(XLSX)

**S3 Table. Strains used in this study.**
(XLSX)

**S4 Table. Oligonucleotides used in this study.**
(XLSX)

**S5 Table. Plasmids used in this study.**
(XLSX)

## Acknowledgments

We thank members of the Zanders lab for comments on the manuscript. We thank the Levin lab for providing Hermes plasmids as well as advice on TN-seq library preparation.

## Author Contributions

**Conceptualization:** R. Blake Billmyre, Michael T. Eickbush, Sarah E. Zanders.

**Data curation:** R. Blake Billmyre, Michael T. Eickbush, Christopher Wood.

**Formal analysis:** R. Blake Billmyre, Michael T. Eickbush, Caroline J. Craig, Jeffrey J. Lange, Christopher Wood.

**Funding acquisition:** Sarah E. Zanders.

**Investigation:** R. Blake Billmyre, Michael T. Eickbush, Caroline J. Craig, Jeffrey J. Lange, Rachel M. Helston.

**Methodology:** R. Blake Billmyre, Michael T. Eickbush, Caroline J. Craig, Jeffrey J. Lange, Christopher Wood, Rachel M. Helston, Sarah E. Zanders.

**Project administration:** R. Blake Billmyre, Sarah E. Zanders.

**Software:** R. Blake Billmyre, Michael T. Eickbush, Jeffrey J. Lange, Christopher Wood.

**Supervision:** R. Blake Billmyre, Sarah E. Zanders.

**Validation:** R. Blake Billmyre, Michael T. Eickbush, Caroline J. Craig, Rachel M. Helston.

**Visualization:** R. Blake Billmyre, Caroline J. Craig, Jeffrey J. Lange, Christopher Wood.

**Writing – original draft:** R. Blake Billmyre.

**Writing – review & editing:** R. Blake Billmyre, Michael T. Eickbush, Caroline J. Craig, Jeffrey J. Lange, Rachel M. Helston, Sarah E. Zanders.

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
