## [Decision Letter · Decision Letter 0]

14 Jul 2022

Dear Dr Billmyre,

Thank you very much for submitting your Research Article entitled 'Genome-wide quantification of contributions to sexual fitness identifies genes required for spore viability and health in fission yeast' to PLOS Genetics.

The manuscript was fully evaluated at the editorial level and by independent peer reviewers. The reviewers appreciated the attention to an important problem, but raised some substantial concerns about the current manuscript. Based on the reviews, we will not be able to accept this version of the manuscript, but we would be willing to review a much-revised version. We cannot, of course, promise publication at that time.

If you decide to revise the manuscript for further consideration at PLOS Genetics, please aim to resubmit within the next 60 days, unless it will take extra time to address the concerns of the reviewers, in which case we would appreciate an expected resubmission date by email to plosgenetics@plos.org.

[LINK]

We are sorry that we cannot be more positive about your manuscript at this stage. Please do not hesitate to contact us if you have any concerns or questions.

Yours sincerely,

Paul J Cullen

Guest Editor

PLOS Genetics

Geraldine Butler

Section Editor: Prokaryotic Genetics

PLOS Genetics

Reviewer's Responses to Questions

**Comments to the Authors:**

Reviewer #1: The manuscript by Billmyre et al. presents a Tn-Seq-based screen to identify genes in S. pombe required for wild-type sexual reproduction. The study identifies 532 such genes. The authors focus additional studies on four genes from the screen: ifs1, plb1, alg9, and sdg1. Deletion phenotypes of plb1, alg9, and sdg1 are unusual. plb1 and alg9 are required for wild-type spore health, as deletion phenotypes for these genes each indicate viable spores with low fitness. The sgd1 gene is reportedly required for growth at low cell densities.

Comments:

In total, the screen appears to have been implemented carefully, and it is nice to see that some phenotypes were confirmed by independent construction of deletion mutants. The remainder of the paper is less clear.

The paper would certainly be strengthened by providing some mechanistic insight regarding plb1, alg9, or sdg1; however, it will probably require significant effort to identify such mechanisms for these phenotypes – particularly for sdg1.

I think it would be helpful to indicate if any transposon insertion bias was observed in the mutagenesis screen from alignment of insertion sites. If findings for this mutagenesis are consistent with previous analysis of the Hermes transposon, it would be fine to reference a previous study.

I don’t have additional comments beyond the ones indicated by previous reviewers of the work.

Reviewer #2: This manuscript from Billmyre et al. describes genome-wide quantification about sexual differentiation of fission yeast using transposon mutagenesis with high-throughput sequencing. This is a very nice piece of work that contains huge amount of data. The results are mostly solid and well presented. Also, the manuscript is well written and will be interesting not only to yeast researchers but also general readers. Therefore, I recommend this manuscript for publication in PLOS genetics.

I have several minor comments as listed below.

In general, "wildtype" is written as "wild type".

Line 334: "Plb1∆" should be "plb1∆".

In the materials and methods section, the notation of the components of the medium is not unified. For example, in line 722, "Adenine" but in line 769, "ade".

Line 782: "ura+" is "ura-"?

Line 884: "glusalase" should be "glusulase".

Figure 3：The notation of "wild type" and "WT" is mixed. It should be unified.

Reviewer #3: Billmyre et al. attempt to globally identify genes ’contributing to sex’ in the fission yeast S. pombe. To this end, they use a Hermes transposon mutagenesis procedure coupled with next-generation sequencing developed by the Levin’s lab. Transposition is induced in a homothallic strain capable of self-mating. The mutagenized population is subjected to media changes to shift from vegetative growth to mating/sporulation and back to vegetative growth. Genes ‘contributing to sex’ are identified as non-essential genes that lack transposon integrations at the end of the regimen. 532 genes are identified in that way, of which 276 have prior ‘sexual fitness‘ annotations according to the authors tally. To get a feeling for how newly identified factors might function, the authors select four of them for more in-depth characterization: ifs1(SPAC3G6.03c), plb1 (SPAC1A6.04c), alg9 (SPAC1834.05), and sdg1 (SPAC12B10.02c).

I enjoyed many aspects of the manuscript. It addresses a biologically important question. The experiments were carefully conducted, varied approaches were used and the manuscript is attractively written. Drawbacks are that evidence is lacking to support that the newly identified genes participate in ‘sex’ (comprising mating-type switching, pheromone sensing, cell fusion, nuclear fusion, meiosis, sporulation, and spore germination) rather than growth. This reduces the general interest of the study as only previously identified genes can be trusted hits. It also limits the value of the gene list as resource to a smaller community. A timeline indicating at which point each gene might be required would have helped, but such a timeline is missing. While well conducted, the detailed analysis of four candidate genes does little to counterbalance these issues.

Could the study function as a useful resource? The approach is to some extent validated by the fact that many genes previously linked to mating/meiosis/spore formation and germination were identified. However, many previously identified genes are also missing, the concordance with previous studies is actually rather low. More worrisome, many of the novel hits are likely false positives. This is because the experimental design does not effectively distinguish genes required for sex from genes required for vegetative growth under the varied conditions used which include a staggering 9 days on MEA. Thus, genes are listed here as ‘important for sex’ while they are, most likely, just important for growth. The authors acknowledge 38 hits ‘putatively’ involved in metabolism (line 270 and Figure S4) but metabolic and biosynthetic pathways are much more prevalent, this can be seen in Table S1. Indeed, the adenine, arginine, asparagine, cysteine, histidine, isoleucine, lysine, proline, methionine, serine, and branched amino acids pathways are all represented. So are lipid metabolism, carbohydrate metabolism, and energy metabolism. But genes required for the biosynthesis of macromolecules are also abundant with numerous ribosomal proteins, ribosome biogenesis factors, translation factors, tRNA modifying enzymes, protein glycosylases and so forth. Altogether, these make up a large portion of the newly identified factors and include many of the factors conserved with vertebrates. Granted, some (and perhaps many) of these and other newly identified factors might be of particular importance for sexual reproduction (as plb1 or alg9 studied here, or gas4 for example which should probably be in the list of known factors), but this will require hard work to determine, one factor at a time, as no attempt was made here to experimentally refine the list in a global manner.

Controls could have been instituted to distinguish ‘genes important for sex’ from ‘genes important for growth’ under the experimental media shifts, the issue cannot be addressed by a posteriori filtering. For instance, a mat1-deleted strain incapable of sex could have been processed in parallel with the h90 wild type, just skipping spore selection by glusulase treatment, to compare the list of genes devoid of transposon integrations in the two strains.

Specific points:

How were the four genes ifs1(SPAC3G6.03c), plb1 (SPAC1A6.04c), alg9 (SPAC1834.05), and sdg1 (SPAC12B10.02c) selected for analysis? Were more genes examined, how were they selected and what was the outcome?

Figure S4 presents a Venn Diagram with 38 ‘annotated metabolic genes’ and 211 genes ‘without relevant annotations’. Could the authors please clarify the definition of ‘relevant’ by providing gene lists (including protein functions and GO terms) for each category in this Venn diagram. This information can be retrieved from PomBase and could perhaps simply be added to Table S1.

Minor points:

Overreaching statement: Line 521 ‘the quorum sensing pathway we have identified here’ is overreaching as no quorum sensing pathway has been identified.

The author should use ‘medium’ rather than ‘media’ when referring to a single medium.

**Have all data underlying the figures and results presented in the manuscript been provided?**

Reviewer #1: Yes

Reviewer #2: Yes

Reviewer #3: **No: **Gene lists including gene products and GO terms should be provided for Figure S4.

PLOS authors have the option to publish the peer review history of their article (what does this mean?). If published, this will include your full peer review and any attached files.

Reviewer #1: No

Reviewer #2: No

Reviewer #3: No

---

## [Decision Letter · Decision Letter 1]

3 Oct 2022

Dear Dr Billmyre,

We are pleased to inform you that your manuscript entitled "Genome-wide quantification of contributions to sexual fitness identifies genes required for spore viability and health in fission yeast" has been editorially accepted for publication in PLOS Genetics. Congratulations!

Yours sincerely,

Paul J Cullen

Guest Editor

PLOS Genetics

Geraldine Butler

Section Editor

PLOS Genetics

Comments from the reviewers (if applicable):

Reviewer's Responses to Questions

**Comments to the Authors:**

Reviewer #1: The authors have addressed my concerns. I don't have any additional comments.

Reviewer #2: The authors have addressed all the issues raised in the previous review.

**Have all data underlying the figures and results presented in the manuscript been provided?**

Reviewer #1: Yes

Reviewer #2: Yes

PLOS authors have the option to publish the peer review history of their article (what does this mean?). If published, this will include your full peer review and any attached files.

Reviewer #1: No

Reviewer #2: No

**Data Deposition**

http://datadryad.org/submit?journalID=pgenetics&manu=PGENETICS-D-22-00625R1

**Press Queries**

---

## [Editor Report · Acceptance letter]

24 Oct 2022

PGENETICS-D-22-00625R1 

Genome-wide quantification of contributions to sexual fitness identifies genes required for spore viability and health in fission yeast 

Dear Dr Billmyre, 

We are pleased to inform you that your manuscript entitled "Genome-wide quantification of contributions to sexual fitness identifies genes required for spore viability and health in fission yeast" has been formally accepted for publication in PLOS Genetics! Your manuscript is now with our production department and you will be notified of the publication date in due course.

With kind regards,

Agnes Pap

PLOS Genetics

On behalf of:
